# ChemEval: A Multi-level and Fine-grained Chemical Capability Evaluation for Large Language Models

**Yuqing Huang**[1][*] **Rongyang Zhang**[1][*] **Xuesong He**[1], **Xuyang Zhi**[1], **Hao Wang**[1][†] **Nuo Chen**[1],
**Zongbo Liu**[1], **Xin Li**[1][†] **Feiyang Xu**[2], **Deguang Liu**[1], **Huadong Liang**[2], **Yi Li**[2], **Jian Cui**[2],
**Yin Xu**[1], **Shijin Wang**[2], **Qi Liu**[1], **Defu Lian**[1], **Guiquan Liu**[1][†] **Enhong Chen**[1][†]
[1]University of Science and Technology of China    [2]iFLYTEK Co., Ltd
{huangyuq,zhangry13}@mail.ustc.edu.cn

## Abstract

The emergence of Large Language Models (LLMs) in chemistry marks a significant advancement in applying artificial intelligence to chemical sciences. While these models show promising potential, their effective application in chemistry demands sophisticated evaluation protocols that address the field's inherent complexities. To bridge this critical gap, we introduce ChemEval, an innovative hierarchical assessment framework specifically designed to evaluate LLMs' capabilities across chemical domains. Our methodology incorporates a distinctive four-tier progression system, spanning from basic chemical concepts to advanced theoretical principles. Sixty-two textual and multimodal tasks are designed to enable researchers to conduct fine-grained analysis of model capabilities and achieve precise evaluation via carefully crafted assessment protocols. The framework integrates carefully curated open-source datasets with expert-validated materials, ensuring both practical relevance and scientific rigor. In our experiments, we evaluated the performance of most main-stream LLMs using both zero-shot and few-shot approaches, with carefully designed examples and prompts. Results indicate that general-purpose LLMs, while proficient in understanding chemical literature and following instructions, struggle with tasks requiring deep chemical expertise. In contrast, chemical LLMs perform better in technical tasks but show limitations in general language processing. These findings highlight both the current limitations and future opportunities for LLMs in chemistry. Our research provides a systematic framework for advancing the application of artificial intelligence in chemical research, potentially facilitating new discoveries in the field.[1]

## 1 Introduction

The advent of large language models has ushered in a transformative era in artificial intelligence, particularly within the domain of natural language processing. The expansive capabilities of these models have not only redefined the boundaries of text generation and understanding (Brown et al., 2020; Ouyang et al., 2022; Touvron et al., 2023; Achiam et al., 2023; Huang et al., 2025; Gu et al., 2025; Liang et al., 2025) but have also opened new avenues for various domains, such as recommendation (Shen et al., 2024) and scientific exploration (Beltagy et al., 2019; Hong et al., 2022; Bhattacharjee et al., 2024). Researchers have adeptly employed LLMs to accelerate the pace of scientific research and instigate a transformative shift in scientific research paradigms. The field of chemistry has notably profited from the integration and advancement of LLMs (Yu et al., 2024; Chen et al., 2024), becoming a key area where these sophisticated technologies have delivered substantial advantages. The intricate nature of chemical research, involving complex molecular interactions and reactions, presents unique challenges that LLMs can address through advanced pattern recognition and predictive analytics.

---

[*]Both authors contributed equally to this research.
[†]Corresponding author.
[1]The code and data are available at https://github.com/USTC-StarTeam/ChemEval.

In order to systematically assess the capabilities of LLMs across various domains and identify areas for their potential enhancement, numerous benchmarking initiatives have been introduced. For instance, the MMLU (Hendrycks et al., 2020) covers 57 tasks spanning basic mathematics, American history, computer science, law, and other fields. The XieZhi (Gu et al., 2024) benchmark includes three major academic categories with 516 specific subjects. However, general benchmarks (Zhong et al., 2023; Huang et al., 2024) often overlook a detailed assessment of chemical knowledge. Although Sun et al. (2024) introduce SciEVAL as a framework for assessing the competencies of LLMs within the scientific domain, the chemistry-related tasks are overly simplistic and do not adequately capture the depth required. Regarding chemistry domain-specific benchmarks, Guo et al. (2023) propose 8 chemical tasks aimed at assessing understanding, reasoning, and explanation abilities, but the benchmark consists of tasks derived from existing public datasets, which may be insufficient to capture the full spectrum of competencies needed for thorough chemical research. Other studies like (White et al., 2023; Liu et al., 2023) have similar problems. Moreover, existing benchmarks fail to address the capability of LLMs to extract chemical information from text and tables. This limitation prevents them from tackling key issues of interest to chemistry researchers and has not fully met the specialized needs of chemistry.

In light of these considerations, we introduce **_ChemEval_**, a benchmark designed to address the gap in the hierarchical assessment framework for LLMs in chemistry by providing a multi-dimensional evaluation. **1). Extensive** tasks are included in ChemEval, which encompasses chemical tasks of interest to researchers that were not included in previous benchmarks. It has four levels, thirteen dimensions, and a total of 62 distinct tasks, covering a vast array of issues within the domain of chemical research. Notably, we innovatively introduce test sets related to information extraction and inductive generation in chemistry. **2). Multimodal** tasks are specifically designed to assess models' capabilities in understanding and reasoning across diverse chemistry-related data types, including text, molecular structure diagrams, and spectral images. **3). Domain experts** in chemistry have meticulously crafted in-depth task datasets and prompts for ChemEval, partly addressing the previous lack of domain-specific data in chemistry benchmarks. Compared to previous work, our study encompasses a broader range of tasks that are of actual concern in chemical research. It assesses models on a graduated scale of capabilities, from general to domain-specific skills, to determine the model's proficiency. Our aim is to construct specialized tasks from the perspective of chemical researchers, thereby providing valuable insights for AI researchers and chemists, and improving large language models' effectiveness in chemical research.

For experiments, we conducted a highly detailed evaluation process, focusing on designing prompts that challenge LLMs, including 0-shot and few-shot settings. We evaluated currently widely used LLMs, including both general LLMs and specialized chemical LLMs, and gained many meaningful insights. This fine-grained evaluation has revealed that though general LLMs excel in Literature Understanding tasks and possess great instruction-following capability, they struggle with tasks that require a deeper understanding of molecular structures and scientific inference. On the other hand, specialized LLMs generally show improved chemical abilities even when their ability to understand literature and instruction-following capability is diminished. This finding underscores the need for significant improvements in the way LLMs are trained and evaluated for chemical tasks. In addition, we explored the impact of few-shot learning and model size on the performance of large language models and provided corresponding insights. We highlight the contributions of this paper as follows:

- We have established an open-source benchmark for LLMs in the field of chemistry, which provides a fine-grained evaluation of their mastery of chemical knowledge as well as their multimodal reasoning capabilities, filling the absence of a holistic benchmark that encompasses the diverse range of tasks within the chemical domain.

- We set up 4 progressive levels and access 13 model capability dimensions through 62 tasks in ChemEval, which is developed through extensive discussions and collaborative design with chemistry researchers, involves constructing novel tasks of interest to chemical researchers and encompasses the primary focal points of chemical research.

- We conducted a fine-grained evaluation of LLMs in chemical tasks, using various prompt settings to assess both general and specialized LLMs. This revealed significant differences between different types of LLMs and identified challenging tasks with potential for optimization. This work offers critical insights to guide researchers in the optimization and application of LLMs, thereby enhancing their effectiveness in chemical research.

## 2  RELATED WORK

**Large Language Models for Chemistry.**  The emergence of Large Language Models (LLMs) has revolutionized Natural Language Processing, with cutting-edge proprietary models like GPT-4o (Hurst et al., 2024) and open-source alternatives such as LlaMA (Touvron et al., 2023) and Qwen (Yang et al., 2024) demonstrating exceptional capabilities across linguistic tasks. However, applying these general models to chemistry reveals significant limitations in domain-specific knowledge. To bridge this gap, researchers have developed specialized approaches: Galactica (Taylor et al., 2022) underwent pre-training on comprehensive scientific corpora, SciGLM (Zhang et al., 2024a) employed strategic fine-tuning with scientific datasets, and ChemCrow (Bran et al., 2023) enhanced performance by integrating expert-designed chemistry tools. Chemistry-focused models, including ChemDFM (Zhao et al., 2024), LlaSMol (Yu et al., 2024), and ChemLLM (Zhang et al., 2024b), incorporate tailored training methodologies, while specialized applications such as Drugchat (Liang et al., 2023) and Drugassist (Ye et al., 2023) specifically address molecular structures and chemical properties. Despite these advancements, achieving comprehensive chemical understanding through LLMs remains a promising frontier for further research and innovation.

**Large Language Models Evaluations for Chemistry.**  The progress made in the field of LLMs is tightly linked to the establishment of robust evaluation frameworks. For general tasks, benchmarks such as MMLU (Hendrycks et al., 2020) and GLUE (Wang et al., 2018) have become standard tools for assessing model capabilities. In the scientific domain, recent initiatives like SciEval (Sun et al., 2024), SceMQA (Liang et al., 2024), and SciAssess (Cai et al., 2024) have been introduced to evaluate scientific reasoning and knowledge. In the chemistry domain, recent benchmarking initiatives such as ChemLLMbench (Guo et al., 2023), ChemBench (Mirza et al., 2024), and MaCBench (Alampara et al., 2025) have emerged, yet each presents significant limitations: ChemLLMbench covers only eight task categories with unreviewed datasets; ChemBench offers 7,000 samples, but is limited by its reliance on multiple-choice questions, lack of open-ended tasks, and insufficient evaluation metrics for chemical experiment design tasks such as synthesis pathway recommendations; while MaCBench introduces multimodal evaluation but exhibits similar constraints in task diversity and assessment metrics. The absence of a precise benchmarking framework impedes LLM advancement in chemistry, a field with complex conceptual knowledge and computational challenges. To address this gap, we introduce ***ChemEval***, a precise evaluation framework designed to rigorously assess LLM capabilities across the multifaceted landscape of chemistry.

## 3  CHEMEVAL

To fill the absence of a holistic benchmark that encompasses the diverse range of tasks within the chemical domain, we introduce a refined benchmark named ***ChemEval*** specifically designed to evaluate the comprehensive capabilities of LLMs within the chemical domain. It not only encompasses text tasks such as literature comprehension and experimental planning, but also incorporates multi-modal tasks, including molecular formula recognition and spectroscopic data analysis. As illustrated in Figure 3.1.1, it contains four levels in the field of chemistry, each of which includes several different chemical dimensions, ensuring a fine-grained evaluation of LLMs. This framework measures the models' ability to understand and infer chemical knowledge from a broad range of dimensions through a series of meticulously designed tasks. In the following subsections, we will provide a detailed introduction to the task content and data construction process of ChemEval.

### 3.1  TASK DESCRIPTION

#### 3.1.1  ADVANCED KNOWLEDGE QUESTION ANSWERING

This segment is pivotal in assessing the models' proficiency in understanding and applying fundamental chemical concepts, which include *Objective Question* dimension and *Subjective Question* dimension, a total of 15 different tasks. Through a blend of objective and subjective tasks, the Advanced Knowledge Question Answering challenges the models to demonstrate their integrated capabilities in areas of chemical terminology, quantitative analysis and cross-modal reasoning. The tasks within this section are designed to be both comprehensive and diagnostic, providing a clear measure of the models' readiness to tackle more advanced chemical inquiries.

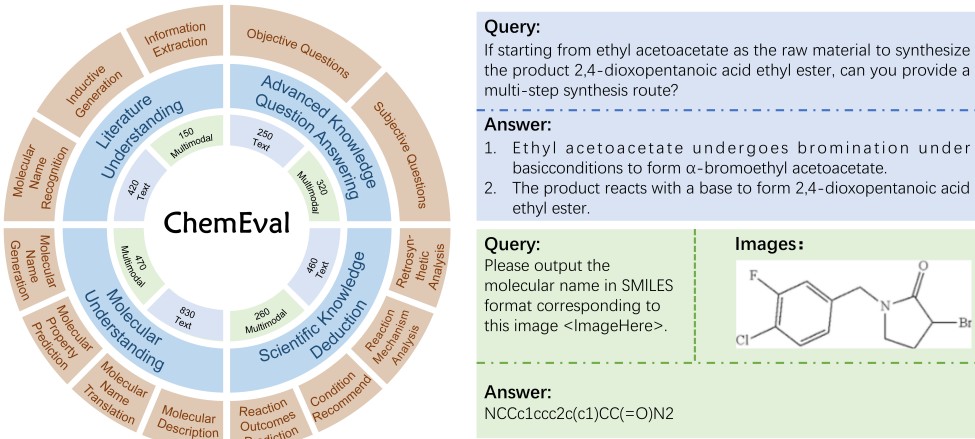

Figure 1: The overview of **_ChemEval_**. It includes 4 progressive levels, evaluating 13 dimensions of LLMs' capabilities and featuring 62 distinct chemical tasks that cover a wide range of chemical knowledge, from foundational concepts to advanced topics suitable for graduate-level research.

### 3.1.2 LITERATURE UNDERSTANDING

Advanced Knowledge Question Answering is designed to assess the model's comprehension and mastery of chemical knowledge, while Literature Understanding evaluates the model's capacity to interpret and assimilate information from chemical literature, which is foundational for subsequent inductive generation tasks. Literature Understanding, which includes the *Inductive Generation*, *Information Extraction*, and *Molecular Name Recognition* dimensions, comprising a total of 19 tasks, delves into tasks crucial for understanding and extracting meaningful information from the chemical literature. The primary focus is on assessing the LLMs' ability to comprehend and extract key information from both textual content and image data in chemical literature, enabling the execution of more complex or information-intensive tasks.

### 3.1.3 MOLECULAR UNDERSTANDING

This section builds upon the previous foundation to assess the model's understanding and generative capabilities at the molecular level. It includes 4 dimensions: *Molecular Name Generation*, *Molecular Name Translation*, *Molecular Property Prediction*, and *Molecular Description*, a total of 15 tasks. Molecular Understanding focuses on core tasks in molecular cognition, aiming to evaluate LLMs in molecular formula conversion, structural diagram interpretation, and the description/prediction of molecular properties based on structural and spectroscopic data. These tasks assess the models' proficiency in interpreting and generating chemical information accurately.

### 3.1.4 SCIENTIFIC KNOWLEDGE DEDUCTION

Having established a solid grasp of basic chemical knowledge, the skill to interpret scientific literature, and the capacity to understand molecular structures, we expect that the model will proceed to conduct deeper chemical reasoning and deduction. The Scientific Knowledge Deduction level covers *Retrosynthetic Analysis*, *Reaction Condition Recommendation*, *Reaction Outcome Prediction* and *Reaction Mechanism Analysis*, a total of 13 tasks, which are essential for effective chemical synthesis. This part evaluates the LLMs' capabilities in retrosynthetic analysis, recommending reaction conditions, predicting reaction outcomes, and analyzing reaction mechanisms. These tasks are essential for efficient chemical synthesis, requiring the model to accurately recognize chemical structures from images and perform complex reasoning and analysis using specific knowledge.

### 3.2 BENCHMARK GENERATION PIPELINE

### 3.2.1 DATA COLLECTION

The overall process of benchmark construction is illustrated in Figure 2. Given the indispensable role of data in the realm of LLMs (Yin et al., 2024), our data collection strategy consists of two primary components: open-source data and domain-expert data.

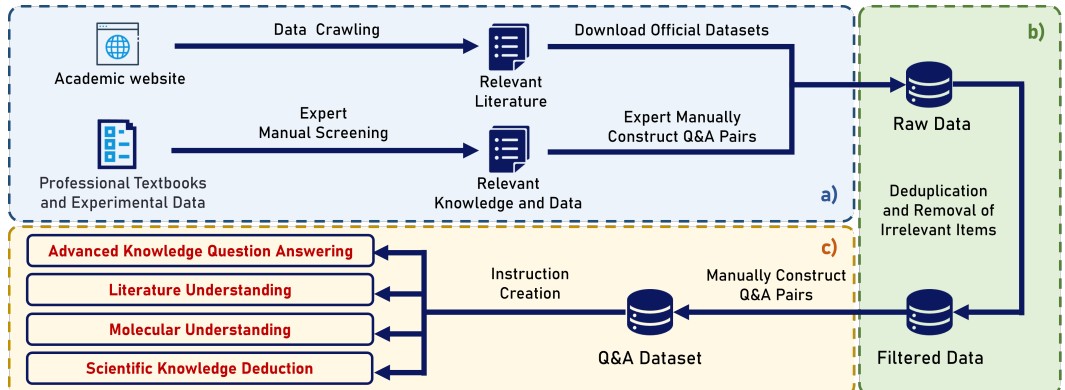

Figure 2: Data collection steps of **ChemEval**. The process is divided into three main steps: a). Data Collection: Raw data is collected from academic websites via web crawling, and experts manually gather data from professional textbooks and experimental data. b). Data Filtering: The raw data undergoes deduplication and removal of irrelevant items to produce filtered data. c). Q&A Pair Construction: Experts manually construct Q&A pairs related to chemistry and create prompt instructions, resulting in four instruction test sets.

For the open-source component, we retrieved relevant publications on chemical language models from academic repositories using keywords such as "chemistry", "large language models", "knowledge question answering", and "information extraction". From these papers, we systematically extracted and codified downstream tasks and their associated datasets to develop our chemical evaluation framework (Yu et al., 2024; Guo et al., 2023; Edwards et al., 2022; Chen et al., 2023; Guo et al., 2021; Zhou et al., 2023; Fang et al., 2023). We then downloaded the official datasets for these tasks, prioritizing those that included an official test set.

However, relying solely on open-source data provides inadequate coverage. To enhance the evaluation's rigor and breadth, we compiled an extensive corpus of domain-expert materials. These raw materials were sourced from: (1) approximately 500 university-level chemistry textbooks, exercise books, and examinations; and (2) around 9,000 real-world experimental records provided by collaborating laboratories. We utilized these resources to manually construct question-answer pairs tailored to specific task types. Crucially, to prevent potential data leakage, materials such as textbook exercises were not directly copied. Instead, chemistry experts used them as references to author novel questions aligned with the target knowledge dimensions and task formats.

### 3.2.2 DATA PROCESSING

Through our data collection efforts, we curated a massive corpus of raw data across the chemical domain. To transform this raw material into a robust benchmarking resource, we executed a rigorous pipeline encompassing selection, filtration, task-specific formatting, and quality assurance.

**Data Filtration and Formatting.** Initially, domain experts conducted a manual filtration process to eliminate unsuitable samples. Data was systematically removed if it fell into any of the following categories: (a) Task-mismatch: data misaligned with our predefined task parameters; (b) Ambiguity: questions susceptible to multiple valid interpretations; (c) Answer multiplicity: tasks permitting several reasonable answers yet lacking a comprehensive standard in the source material; (d) Outdated information: content relying on obsolete or revised chemical facts; and (e) Redundancy: duplicated or highly similar items. This strict protocol resulted in the exclusion of approximately 200 items (roughly 2% of the initial pool). Upon passing this stage, the data was processed across three distinct dimensions:

*(1) Advanced knowledge question answering.* We formulated question-answer pairs using collegiate and graduate-level textbooks. This set spans seven core disciplines (organic, inorganic, materials, analytical, biochemistry, physical, and polymer chemistry).

*(2) Literature understanding.* We extracted pertinent excerpts from scientific publications, pairing them with meticulously designed questions and reference answers.

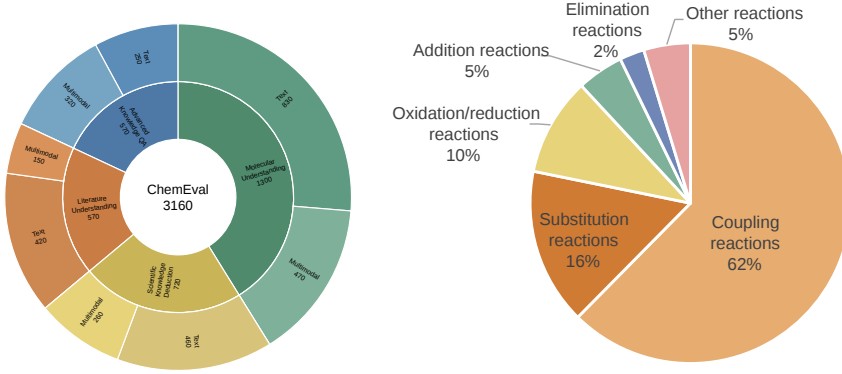

(a) Distribution of evaluation tasks in ChemEval.     (b) Percentage distribution of reaction types in the ChemEval.

Figure 3: Distribution of the ChemEval benchmark dataset. (a) A hierarchical breakdown of evaluation tasks categorized by four core capability levels and modalities. (b) The percentage distribution of major chemical reaction types evaluated within the dataset.

*(3) Molecular understanding and scientific reasoning.* By merging open-source datasets with proprietary empirical records from our partner laboratories, we built evaluation sets that rigorously simulate domain-specific scientific scenarios.

**Quality Assurance and De-contamination.** To uphold the integrity of this dataset, we enforced a strict three-tier quality assurance pipeline: Annotation, Review, and Final Audit. First, undergraduate chemistry students, guided by a standardized SOP, conducted the primary annotations. Second, a graduate student team cross-examined these annotations to verify logical consistency. Finally, chemistry faculty members executed a conclusive audit, ensuring all ground-truth answers were strictly fact-based. Crucially, to preclude any risk of data leakage, we rigorously cross-referenced our downstream test sets against the training corpora of existing open-source chemical models, systematically purging any overlapping instances.

### 3.2.3 DATA STATISTICS

Following the comprehensive data processing and rigorous deduplication pipeline described above, we successfully finalized the ChemEval benchmark dataset. As illustrated in Figure 3(a), this extensive curation effort yielded a total of 3,160 high-quality evaluation instances. To ensure that our benchmark captures both the breadth and depth required to assess state-of-the-art large language models, we strategically combined established evaluation paradigms with a substantial volume of novel, challenging tasks. In total, ChemEval encompasses 62 distinct task categories. Among these, 25 tasks were carefully adapted from highly regarded open-source repositories to maintain alignment with existing community standards. The remaining 37 proprietary tasks were engineered entirely in-house by our domain experts to specifically address complex chemical reasoning capabilities that have been previously overlooked by existing benchmarks.

To comprehensively probe the multifaceted capabilities of modern AI models, these evaluation instances are systematically structured into distinct text and multimodal subsets. The text-only subset contains 1,960 meticulously annotated examples, comprising 18 open-source tasks and 24 in-house designed tasks. The multimodal subset consists of 1,200 test examples, which integrate 12 open-source tasks with 30 in-house designed visual-textual tasks. Notably, several core tasks are intentionally formulated in both text and multimodal formats, establishing a robust framework to explicitly evaluate and compare the cross-modal alignment capabilities of current vision-language models.

Furthermore, to provide a rigorous and chemically meaningful testing ground for scientific reasoning, the dataset was designed to offer extensive coverage of major chemical reaction mechanisms frequently encountered in real-world research and industrial applications. As summarized in Figure 3(b), coupling reactions account for the largest proportion of the data. This is followed by fundamental reaction classes including substitution, oxidation/reduction, addition, and elimination reactions. The remaining instances are grouped into an "other" category that covers 9 distinct and highly specialized mechanisms, such as rearrangement, hydrolysis, and cyclization, which often test

a model's grasp of complex reaction pathways and structural transformations. This diverse distribution ensures that LLMs are evaluated not merely on rote memorization, but on their genuine ability to generalize across a wide spectrum of chemical transformations. Complete statistics regarding the precise data distribution and task provenance are detailed in Appendix B.

### 3.2.4 INSTRUCTION CREATION

To evaluate the effectiveness of the model, we constructed task-specific prompts and 3-shot task-specific prompts for text downstream tasks (Wei et al., 2022). For downstream tasks with open-source datasets, to facilitate evaluation, the evaluation system in this paper strengthens the format of the output data based on its instructions. For the domain expert-built part, the evaluation system in this paper will design instructions for task introduction and formatted output according to the task type, and continuously adjust the instructions based on the return results of GPT-4o, thereby strengthening the instructions for different self-constructed downstream tasks.

### 3.2.5 METRICS

In this study, we utilize a range of evaluation metrics for a fine-grained assessment of LLMs' performance across diverse tasks. For the majority of tasks, we utilize the F1 score and Accuracy. In addition, we utilize BLEU (Papineni et al., 2002), Exact Match, Normalized Root Mean Square Error, Valid Output Ratio, LLMs Score, L2 Score, and Overlap as evaluation metrics for different tasks to accommodate various task requirements. A detailed introduction to the metrics is provided in the appendix C.2.

## 4 EXPERIMENT

### 4.1 SETUP

To conduct a fine-grained diagnostic evaluation of LLMs' chemical capabilities, our framework assesses both general and specialized models. For general LLMs, we include OpenAI-o1/o3-mini (Jaech et al., 2024), GPT-4o (Hurst et al., 2024), Claude-3.7-Sonnet (Anthropic, 2025), Gemini-2.5-pro (Team et al., 2023), Qwen2.5-7B/14B/32B/72B (Yang et al., 2024), LLaMA3.3-8B (Touvron et al., 2023), Grok3 (xAI, 2025), and DeepSeek-V3/R1 (Liu et al., 2024). For chemistry-specific LLMs, we evaluate ChemDFM (Zhao et al., 2024), LlaSMol (Yu et al., 2024), ChemLLM (Zhang et al., 2024b) and ChemSpark[2]. For multimodal chemical tasks, we evaluated mainstream MLLMs, including GPT-4o (Hurst et al., 2024), Claude-3.7-Sonnet (Anthropic, 2025), Qwen-VL Max (Bai et al., 2023), Phi-Vision-3.5 (Abdin et al., 2024), across four levels of multimodal chemistry tasks. We used the official APIs of general models for evaluation and ran the chemistry-specific models on two A40 48GB GPUs. We employed greedy decoding for all LLM inference in our experiments.

To illustrate the capability of LLMs in various chemical tasks, we present their average zero-shot performance across four levels, with detailed results shown in the table 1. To assess their adaptability and in-context learning abilities, we also report three-shot performance across the same levels. Some tasks, such as *Chemical Paper Abstract Generation*, are not included in our three-shot evaluation due to context length limitations.

### 4.2 PERFORMANCE RESULTS

We evaluate the models' performance for each task across four assessment Levels. Evaluation results for text tasks are summarized in Table 1, and those for multimodal tasks are summarized in Table 9. Certain models are unable to address specific tasks entirely. For example, LLaMA3.3-8B demonstrates poor instruction-following capabilities in TempRec task in the 0-shot setting, which significantly impairs its ability to generate responses based on task prompts. Consequently, we are unable to provide numerical results for the tasks affected by this limitation. We further discuss the key findings of our benchmark and analyze how different LLM configurations influence performance, offering practical insights for the development of chemistry-specific benchmarks.

---

[2]https://www.modelscope.cn/models/iflytek/Spark-Chemistry-X1-13B

Table 1: Performance overview of representative multi-level 0-Shot text tasks on ChemEval. Claude3.7T denotes Claude 3.7-Sonnet-Thinking, whereas Claude3.7N denotes Claude 3.7-Sonnet. For the complete experimental results, please refer to the appendix D.1.

| Dimension | Task | Metric | OpenAI-o1 | GPT-4o | Claude3.7T | Deepseek-R1 | Deepseek-V3 | Qwen2.5-72B | Llama3.3-8B | Gemini-2.5-Pro | ChemDFM | ChemLLM | LlaSMol | ChemSpark | Chemcrow |
|---|---|---|---|---|---|---|---|---|---|---|---|---|---|---|---|
| *Advanced Knowledge Question Answering* | | | | | | | | | | | | | | | |
| ObjQA | MCTask | Accuracy | 74.00 | 66.80 | 62.80 | 82.40 | 76.00 | 67.20 | 40.40 | 87.60 | 41.20 | 24.40 | 24.00 | 43.60 | 58.00 |
| ObjQA | FBTask | LLM Score | 60.92 | 51.19 | 45.28 | 59.41 | 63.88 | 53.92 | 34.17 | 63.95 | 24.16 | 34.97 | 13.92 | 24.57 | 43.14 |
| ObjQA | TFTask | Accuracy | 46.00 | 57.60 | 58.80 | 75.20 | 67.20 | 58.40 | 46.00 | 77.60 | 46.00 | 19.20 | 58.00 | 50.00 | 74.00 |
| SubjQA | SATask | LLM Score | 64.50 | 61.20 | 56.70 | 68.50 | 71.70 | 58.50 | 38.40 | 72.00 | 32.20 | 13.20 | 14.50 | 33.60 | 43.50 |
| SubjQA | CalcTask | LLM Score | 78.00 | 61.80 | 55.74 | 76.10 | 79.20 | 61.90 | 28.00 | 82.40 | 14.70 | 15.90 | 7.50 | 18.50 | 43.50 |
| *Literature Understanding* | | | | | | | | | | | | | | | |
| InfoE | CNER | F1 | 64.56 | 65.76 | 60.21 | 64.14 | 60.85 | 61.61 | 55.34 | 68.30 | 41.17 | 0.16 | 11.62 | 71.44 | 57.46 |
| InfoE | CERC | F1 | 22.37 | 25.66 | 25.19 | 27.18 | 24.94 | 26.05 | 17.31 | 25.43 | 8.74 | 0.24 | 1.24 | 39.27 | 22.05 |
| InfoE | SubE | Accuracy | 73.71 | 66.32 | 61.59 | 75.18 | 61.26 | 62.56 | 64.02 | 72.05 | 20.07 | 0.00 | 0.00 | 74.38 | 50.91 |
| InfoE | AddE | F1 | 81.67 | 85.00 | 79.33 | 82.67 | 80.67 | 84.00 | 45.81 | 95.00 | 45.00 | 0.00 | 0.00 | 65.00 | 43.33 |
| InfoE | SolvE | F1 | 86.50 | 85.00 | 87.60 | 90.20 | 88.50 | 85.00 | 75.47 | 83.17 | 80.50 | 1.67 | 0.00 | 83.79 | 87.50 |
| InfoE | TempE | F1 | 70.00 | 67.00 | 72.00 | 65.00 | 72.00 | 65.00 | 62.00 | 69.00 | 74.33 | 3.23 | 0.00 | 83.00 | 65.00 |
| InfoE | TimeE | F1 | 95.00 | 95.00 | 95.00 | 95.00 | 95.00 | 90.00 | 90.00 | 94.00 | 78.00 | 23.10 | 25.00 | 95.00 | 95.00 |
| InfoE | ProdE | Accuracy | 90.25 | 86.09 | 82.39 | 91.20 | 87.52 | 84.86 | 74.54 | 92.82 | 34.73 | 0.00 | 0.00 | 94.40 | 71.38 |
| InfoE | CharME | F1 | 51.67 | 72.85 | 81.01 | 21.33 | 81.80 | 74.57 | 44.18 | 73.11 | 27.26 | 0.00 | 0.00 | 12.98 | 25.00 |
| InfoE | CatTE | F1 | 95.00 | 94.00 | 82.00 | 99.00 | 100.00 | 100.00 | 65.00 | 96.00 | 49.00 | 0.00 | 5.00 | 31.00 | 85.00 |
| InfoE | YieldE | F1 | 85.00 | 79.00 | 61.00 | 77.70 | 65.00 | 65.00 | 46.00 | 74.00 | 45.00 | 0.00 | 5.00 | 61.00 | 50.00 |
| InducGen | AbsGen | LLM Score | 63.75 | 63.00 | 63.00 | 65.00 | 64.75 | 64.75 | 62.00 | 67.25 | 0.00 | 5.50 | 26.25 | 38.25 | 57.50 |
| InducGen | OLGen | LLM Score | 25.00 | 35.50 | 26.50 | 37.00 | 27.00 | 24.25 | 22.75 | 39.50 | 0.00 | 3.75 | 31.25 | 30.50 | 32.50 |
| InducGen | TopC | Accuracy | 55.00 | 49.00 | 56.00 | 57.00 | 50.00 | 64.00 | 32.00 | 67.00 | 51.00 | 0.00 | 0.00 | 30.00 | 45.00 |
| InducGen | ReactTR | F1 | 25.00 | 32.00 | 29.00 | 21.00 | 28.00 | 22.00 | 26.00 | 31.00 | 13.00 | 0.00 | 5.00 | 17.00 | 5.00 |
| *Molecular Understanding* | | | | | | | | | | | | | | | |
| MNGen | MolNG | Tanimoto (valid) | 49.80 (72%) | 39.30 (89%) | 33.85 (70%) | 56.05 (87%) | 51.19 (96%) | 20.58 (79%) | 5.83 (40%) | 71.11 (93%) | 47.06 (69%) | 0.00 (0%) | 3.71 (76%) | 74.81 (98%) | 40.92 (90%) |
| MNTrans | IUPAC2MF | L2 | 0.7737 | 0.5304 | 0.3252 | 0.6026 | 0.6176 | 0.3407 | 0.2433 | 0.8382 | 0.6119 | 0.0454 | 0.0000 | 0.8807 | 0.1408 |
| MNTrans | SMILES2MF | L2 | 0.6330 | 0.3627 | 0.3618 | 0.4402 | 0.3563 | 0.2848 | 0.1728 | 0.6574 | 0.6399 | 0.0375 | 0.0000 | 0.8133 | 0.3089 |
| MNTrans | IUPAC2SMILES | Tanimoto (valid) | 29.72 (50%) | 34.71 (83%) | 31.89 (68%) | 30.70 (63%) | 46.07 (88%) | 15.90 (76%) | 5.24 (30%) | 61.35 (87%) | 46.71 (88%) | 0.00 (100%) | 4.70 (56%) | 87.84 (100%) | 25.68 (64%) |
| MNTrans | SMILES2IUPAC | Exact Match | 0.00 | 0.00 | 0.00 | 1.20 | 0.00 | 0.00 | 0.00 | 1.20 | 0.00 | 0.00 | 0.00 | 14.00 | 0.00 |
| MNTrans | SMILES2IUPAC | BLEU | 3.24 | 0.96 | 3.27 | 4.17 | 1.67 | 0.33 | 0.44 | 13.55 | 0.56 | 0.00 | 0.00 | 48.25 | 0.38 |
| MNTrans | SMILES2IUPAC | Tanimoto | 0.00 | 12.08 | 22.73 | 25.90 | 19.16 | 13.01 | 3.71 | 56.82 | 2.06 | 0.00 | 2.22 | 66.26 | 0.00 |
| MNTrans | S2S | Tanimoto (valid) | 9.72 (42%) | 13.41 (62%) | 9.37 (40%) | 16.04 (71%) | 16.27 (62%) | 11.47 (50%) | 1.74 (12%) | 13.13 (44%) | 2.12 (25%) | 0.00 (50%) | 0.60 (48%) | 87.36 (94%) | 9.83 (38%) |
| MPP | MolPC | Accuracy | 67.50 | 64.57 | 58.90 | 53.54 | 48.73 | 48.13 | 47.26 | 63.63 | 61.35 | 0.00 | 46.50 | 85.57 | 46.00 |
| MPP | MolPR | NRMSE (valid) | 12.3852 (99%) | 9.9322 (51%) | 13.9702 (92%) | 15.8881 (100%) | 8.3675 (98) | 13.0756 (100%) | 61.4736 (62%) | 11.7270 (100%) | 394.9424 (83%) | 179.3606 (93%) | 29.9686 (73%) | 1.2142 (100%) | 0.3408 (38%) |
| MolDesc | Mol2PC | LLM Score | 19.00 | 7.00 | 9.80 | 11.90 | 13.50 | 20.80 | 2.10 | 0.70 | 3.10 | 0.30 | 0.00 | 48.90 | 21.00 |
| *Scientific Knowledge Deduction* | | | | | | | | | | | | | | | |
| ReSyn | SubRec | F1 | 1.00 | 0.00 | 1.46 | 1.63 | 2.27 | 1.06 | 0.27 | 0.00 | 3.99 | 0.00 | 0.00 | 12.37 | 0.00 |
| ReSyn | PathRec | LLM Score | 30.63 | 22.88 | 0.36 | 52.75 | 37.38 | 41.13 | 20.88 | 43.75 | 24.13 | 10.88 | 10.00 | 38.75 | 48.75 |
| ReSyn | SynDE | NRMSE (valid) | - (5%) | - (0%) | - (0%) | - (0%) | - (0%) | 0.2670 (100%) | - (0%) | - (0%) | - (0%) | 33.0049 (78%) | 1.2374 (45%) | 1.7992 (87%) | - (0%) |
| RCRec | LRec | F1 | 0.00 | 13.20 | 2.00 | 6.80 | 7.60 | 4.40 | 2.13 | 0.00 | 26.00 | 0.00 | 0.00 | 37.60 | 18.00 |
| RCRec | RRec | F1 | 25.64 | 15.80 | 27.43 | 21.93 | 8.35 | 37.75 | 8.78 | 0.73 | 13.13 | 0.00 | 0.50 | 63.72 | 36.65 |
| RCRec | SolvRec | F1 | 10.00 | 20.40 | 18.80 | 22.40 | 24.00 | 50.40 | 3.63 | 0.00 | 10.53 | 0.00 | 0.50 | 30.40 | 12.00 |
| RCRec | CatRec | F1 | 0.00 | 0.00 | 0.00 | 0.00 | 0.00 | 0.00 | 0.00 | 0.00 | 0.00 | 0.00 | 0.00 | 0.50 | 0.00 |
| RCRec | TempRec | NRMSE (valid) | 0.3278 (100%) | 0.2545 (100%) | 0.2263 (100%) | 0.2078 (100%) | 0.2096 (100%) | 0.3782 (100%) | - (0%) | 0.1814 (100%) | 0.3811 (99%) | 1.1184 (98%) | 0.8658 (100%) | 0.2742 (100%) | 0.2392 (85%) |
| RCRec | TimeRec | NRMSE (valid) | 0.2746 (100%) | 0.2468 (100%) | 0.3662 (100%) | 0.2291 (100%) | 0.2579 (100%) | 0.2022 (100%) | - (0%) | 0.2425 (100%) | 0.4732 (100%) | 1.7937 (98%) | 0.4351 (80%) | 0.3937 (100%) | 0.5209 (70%) |
| ROP | PPred | F1 | 21.33 | 1.67 | 12.27 | 11.97 | 0.93 | 1.73 | 0.00 | 29.20 | 18.80 | 0.00 | 16.00 | 56.40 | 0.00 |
| ROP | YPred | Accuracy | 12.00 | 43.50 | 16.00 | 11.00 | 22.50 | 26.00 | 35.50 | 17.50 | 7.20 | 0.00 | 28.00 | 72.00 | 24.00 |
| ROP | RatePred | Overlap | 21.08 | 13.81 | 9.06 | 17.12 | 17.71 | 10.71 | 6.92 | 27.01 | 3.79 | 0.00 | 3.68 | 2.90 | 0.00 |
| RMA | IMDer | LLM Score | 80.00 | 81.50 | 81.50 | 79.50 | 80.50 | 77.25 | 81.25 | 82.25 | 76.00 | 4.75 | 1.50 | 92.75 | 28.75 |

### 4.2.1 THE MODELS' PERFORMANCE ACROSS FOUR LEVELS.

The performance comparison of LLMs across four levels reveals distinct strengths and weaknesses:

**Basic Knowledge.** Within the level of Advanced Knowledge Question Answering, the results reveal that OpenAI-o1 exhibits superior performance in objective questions, and Gemini outperforms other models in subjective questions, which indicates the importance of reasoning ability in Q&A questions. Additionally, general LLMs like GPT-4o and Qwen2.5-72B also perform well in literature understanding. However, chemistry-specialized models (except ChemSpark) struggle with general tasks, highlighting instruction fine-tuning challenges, which suggests that general LLMs succeed stem from superior document comprehension and reasoning abilities.

**Chemical Expertise.** As for Molecular Understanding, ChemSpark stands out in these tasks demanding an in-depth grasp of chemical molecules. Most models perform poorly in molecular name translation due to a lack of formatting constraints in their outputs, owing to the complexity of molecular expressions. ChemSpark's advantage be attributed to training on diverse chemical literature with various molecular formula formats. Besides, we observed that when confronted with complex tasks requiring quantitative calculations, models tend to provide overly cautious responses, such as "quantification software (Gaussian, ORCA, etc.) is needed" or "cannot determine from a 2D structure", which significantly reduces the practical value of their answers.

**Chemistry-specialized LLMs.** Compared to general LLMs, specialized chemistry models show distinct patterns: *1). Drawbacks:* Chemical LLMs notably lower in advanced knowledge answering and literature comprehension, suggesting catastrophic forgetting during fine-tuning compromises their foundational language processing capabilities. *2). Advantages:* Chemical models excel in tasks requiring specialized terminology and molecular properties. General models perform adequately on simpler tasks but struggle with complex chemical knowledge processing and inference. *3). Instruction-following ability:* Chemistry-specific LLMs demonstrate significantly lower instruction-following capability than general LLMs, likely due to limited exposure to diverse tasks during training. Without output format restrictions, these models default to patterns matching their fine-tuning data, sometimes producing interpretable results where format-constrained prompts are removed, though with uncertain accuracy. This instruction-following deficiency could significantly impact the practical utility of these specialized models despite their domain expertise.

Table 2: 3-shot performance changes on text tasks relative to 0-shot in ChemEval. The symbols and accompanying values show performance changes compared to 0-shot, where '↑' indicates an increase, '↓' a decrease, and '-' no change. The three values in the last column (↑, ~, ↓) represent the number of tasks that show a significant increase, remain unchanged, and significantly decrease.

| Task Metric | SATask LLM Score | CalcTask LLM Score | SubE Accuracy | TempE F1 | ProdE Accuracy | ReactTR F1 | MolPC Accuracy | LRec F1 | PathRec LLM Score | RatePred Overlap | Change (↑, ~, ↓) |
|---|---|---|---|---|---|---|---|---|---|---|---|
| OpenAI-o1 | 68.50 ↑4.00 | 78.50 ↑0.50 | 78.01 ↑4.30 | 75.00 ↑5.00 | 91.48 ↑1.23 | 60.00 ↑35.00 | 71.60 ↑4.10 | 18.00 ↑18.00 | 40.63 ↑10.01 | 14.41 ↓6.67 | (9, 0, 1) |
| GPT-4o | 61.00 ↓0.20 | 59.10 ↓2.70 | 65.93 ↓0.39 | 73.00 ↑6.00 | 86.88 ↑0.79 | 71.00 ↑39.00 | 68.55 ↑3.98 | 15.60 ↑2.40 | 25.00 ↑2.13 | 20.27 ↑6.47 | (7, 0, 3) |
| Gemini-2.5-Pro | 70.00 ↑2.00 | 81.60 ↓0.80 | 76.29 ↑4.24 | 77.00 ↑8.00 | 93.75 ↑0.93 | 59.00 ↑28.00 | 67.62 ↑3.99 | 0.00 | 43.00 ↓0.75 | 29.08 ↑2.06 | (6, 1, 3) |
| Deepseek-v3 | 70.40 ↓1.30 | 77.40 ↓1.80 | 75.78 ↑14.51 | 80.00 ↑8.00 | 91.75 ↑4.23 | 46.00 ↑18.00 | 55.79 ↑7.06 | 11.60 ↑4.00 | 24.00 ↓13.38 | 13.45 ↓4.26 | (6, 0, 4) |
| Qwen2.5-72B | 60.80 ↑2.30 | 61.61 ↓0.29 | 70.10 ↑7.54 | 80.00 ↑15.00 | 84.05 ↓0.81 | 61.00 ↑39.00 | 56.87 ↑8.74 | 16.40 ↑12.00 | 33.38 ↓7.75 | 15.82 ↑5.10 | (7, 0, 3) |
| Llama3.3-8B | 29.00 ↓9.40 | 19.70 ↓8.30 | 57.71 ↓6.31 | 69.00 ↑7.00 | 73.26 ↓1.28 | 39.00 ↑13.00 | 53.20 ↑5.95 | 2.40 ↑0.27 | 17.88 ↓3.00 | 14.29 ↑7.38 | (5, 0, 5) |
| ChemDFM | 30.50 ↓1.70 | 16.40 ↑1.70 | 20.04 ↓0.03 | 41.00 ↓33.33 | 8.83 ↓25.90 | 26.00 ↑13.00 | 56.65 ↓4.70 | 12.49 ↓13.51 | 28.75 ↑4.63 | 17.46 ↑13.67 | (4, 0, 6) |
| ChemLLM | 11.50 ↓1.70 | 35.46 ↑19.56 | 0.00 | 1.53 ↓1.70 | 0.00 | 0.00 | 0.00 | 0.00 | 6.75 ↓4.13 | 0.00 | (1, 6, 3) |
| LlaSMol | 23.50 ↑9.00 | 68.37 ↑60.87 | 0.00 | 0.00 | 0.00 | 0.00 ↓5.00 | 40.00 ↓6.50 | 0.00 | 17.50 ↑7.50 | 0.00 ↓3.68 | (3, 4, 3) |
| ChemSpark | 31.60 ↓2.00 | 15.80 ↓2.70 | 72.86 ↓1.52 | 80.00 ↓3.00 | 98.40 ↑4.00 | 32.00 ↑15.00 | 82.88 ↓2.68 | 16.80 ↓20.80 | 27.00 ↓11.75 | 11.03 ↑8.13 | (3, 0, 7) |

### 4.2.2 FACTORS AFFECTING MODEL PERFORMANCE IN CHEMISTRY TASKS

**The influence of few-shot.** Our experiment results of ICL are shown in Table 2. Few-shot prompting influence model performance across different tasks. General LLMs tended to benefit from few-shot examples, especially in subjective question answering and literature understanding. In contrast, specialized chemistry models often show performance decreases with few-shot prompting, which may be attributed to catastrophic forgetting of ICL capabilities during task-specific fine-tuning. For complex chemistry-specific tasks, performance variations remain minimal across all models, reflecting the inherent difficulty of these tasks and limitations in capturing expert-level chemical reasoning.

**The impact of model scaling.** We conducted experiments on Qwen2.5 models of different sizes. The results, as shown in Table 3, indicate a trend that increasing model size correlates with improves performance in most tasks, with notable gains in advanced knowledge Q&A and literature understanding. However, molecular understanding and scientific knowledge deduction tasks show minimal improvement as the model scales. Tasks requiring specialized chemical knowledge (e.g., IUPAC2SMILES, CatRec) remain challenging despite parameter increases, with some tasks like MolPC even showing performance declines. This suggests that model scaling alone is insufficient for complex chemical tasks without specialized training data.

**The impact of thinking models.** While intuitively it may seem that thinking models possess stronger reasoning capabilities and might benefit in complex chemical tasks, our experimental comparison of OpenAI-o1 versus GPT-4o and DeepSeek-R1 versus DeepSeek-V3 reveals a more nuanced reality. Although thinking models occasionally excel in specific tasks such as reaction product prediction, they demonstrate comparable performance to general models across most chemical tasks, with each architecture exhibiting distinct strengths in different tasks. Additionally, when prompted to employ chain-of-thought reasoning, some models declined to respond to certain tasks, citing insufficient information to formulate complete answers. We consider that the primary limitation in addressing sophisticated chemical challenges lies not in long reasoning ability but rather in insufficient domain-specific knowledge.

**Stability analysis.** As illustrated in the Table 12, we conducted robustness testing on multiple models and analyzed the stability of metrics across various tasks in the benchmark. The results demonstrate that the standard deviation for the vast majority of metrics does not exceed 5.0, indicating consistent performance across evaluations. These results collectively indicate that our evaluation framework is robust, providing consistent and reliable assessments of system performance.

### 4.2.3 MULTIMODAL CHEMISTRY TASKS

The Table 9 illustrates the performance of mainstream multimodal large language models on ChemEval's multimodal tasks. Entries marked as '-' indicate instances where models failed to generate meaningful responses. Examining results across both Advanced Knowledge QA and Literature Understanding levels reveals that while most models demonstrate satisfactory performance on elementary tasks such as molecular formula identification, they exhibit notable limitations when confronted with more sophisticated challenges involving chemical reaction pathways or molecular properties, as evidenced in Pathway Parsing and Multiple Choice tasks. The performance degra-

Table 3: The impact of model scaling on task performance.

| Task Metric | MCTask Accuracy | SATask LLM Score | CalcTask LLM Score | CharME F1 | CatTE F1 | MolPC Accuracy | CatRec F1 | PPred F1 | YPred Accuracy |
|---|---|---|---|---|---|---|---|---|---|
| Qwen2.5-7B | 59.60 | 50.80 | 43.60 | 43.00 | 64.00 | 64.04 | 0.00 | 0.00 | 67.00 |
| Qwen2.5-14B | 64.80 | 57.20 | 50.80 | 67.92 | 75.00 | 64.22 | 0.00 | 0.00 | 33.50 |
| Qwen2.5-32B | **67.20** | 58.10 | 57.40 | **79.42** | **100.00** | **67.70** | 0.00 | 0.53 | **85.00** |
| Qwen2.5-72B | **67.20** | **58.50** | **61.90** | 74.57 | **100.00** | 48.13 | 0.00 | **1.73** | 26.00 |

dation becomes even more pronounced in Molecular Understanding and Scientific Knowledge Deduction tasks, where models demonstrate considerable difficulty. These advanced tasks present a multifaceted challenge, requiring models to accurately recognize molecular structures and reaction equations from visual inputs while leveraging comprehensive chemical domain knowledge to formulate correct responses. This combination presents a considerable challenge to the models' integrated capabilities. It is worth noting that our evaluation exclusively assessed general-purpose multimodal large language models, without including specialized multimodal models designed specifically for chemical applications. Given that multimodal capabilities are increasingly crucial in chemical research, we think of this as a critical area that warrants further investigation and development.

## 5 LIMITATIONS AND FUTURE WORKS

Although ChemEval addresses a critical gap in evaluating LLMs in the chemistry domain by providing a comprehensive reference for model capability assessment, several limitations within the benchmark design remain. First, to achieve broad task diversity within finite resource constraints, the number of evaluation instances per individual task is relatively limited. While this design allows ChemEval to serve as a highly effective diagnostic tool for probing multifaceted capabilities, it may not suffice for exhaustive, large-scale stress testing of specific narrow tasks. Second, the current benchmark predominantly focuses on static, single-turn evaluation formats. It does not yet incorporate dynamic, interactive environments or systematically assess models' abilities to utilize external professional chemical tools and execute complex agentic workflows, which is essential for evaluating real-world scientific workflows. For the future development of ChemEval, we aim to expand the scale of individual task datasets and introduce dynamic, agent-based evaluation protocols to assess end-to-end tool-use capabilities in advanced chemical research. Additionally, we plan to involve chemical experts in manually evaluating complex LLM outputs to improve evaluation reliability, safety oversight, and alignment with human judgment.

## 6 CONCLUSION

In this paper, we presented ChemEval, a fine-grained benchmark for evaluating LLMs on chemical tasks across four assessment levels. Our experiments show that while general-purpose LLMs excel in literature understanding and benefit from scaling and few-shot prompting, they struggle with molecular understanding and scientific knowledge inference. Chemistry-specialized models exhibit advantages in terminology and molecular property tasks but face challenges such as catastrophic forgetting and weaker instruction-following ability. These findings highlight that improvements in parameter scaling or reasoning depth alone are insufficient to address complex chemical tasks. Instead, progress requires tighter integration of LLMs with domain-specific knowledge, chemical simulation tools, and multimodal data. We hope ChemEval provides both a rigorous evaluation framework and a foundation to inspire the development of chemistry-aware LLMs, ultimately driving advances in chemical research and accelerating the integration of artificial intelligence into the natural sciences.

## ACKNOWLEDGMENTS

This work was supported by the Major Project of the National Social Science Fund of China: "Key Enabling Technologies for Agentic Document and Information Services in the Context of Digital and Intelligent Transformation" (Project No. 23&ZD228), the National Natural Science Foundation of China (Nos. 62441239, U23A20319, 62441227, and 62472394), and the Anhui Province Science and Technology Innovation Project (Nos. 202423k09020010 and 202423k09020011). We gratefully acknowledge the USTC Supercomputing Center for providing the computational resources essential to this research.

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

## A CHEMEVAL TASKS

In order to systematically evaluate the multifaceted capabilities of large language models in the domain of chemistry, we propose a multi-level and fine-grained evaluation framework that encompasses a broad spectrum of chemical knowledge and reasoning tasks. This framework is delineated into four primary categories: Advanced Knowledge Question Answering, Literature Understanding, Molecular Understanding, and Scientific Knowledge Deduction. Each of these categories represents a progressively sophisticated level of chemical problem-solving, ranging from the assessment of fundamental chemical concepts and literature comprehension to molecular-level reasoning and high-level scientific deduction. The constituent tasks within each category are meticulously designed to interrogate specific competencies, such as objective and subjective answering, information extraction, inductive generation, molecular property prediction, and retrosynthetic analysis. Collectively, this comprehensive benchmark offers a granular and holistic evaluation of LLMs' proficiency in both the understanding and application of chemical knowledge, thereby illuminating their potential utility and limitations in diverse chemical informatics applications.

### A.1 ADVANCED KNOWLEDGE QUESTION ANSWERING

This segment is pivotal in assessing the models' proficiency in understanding and applying fundamental chemical concepts, which include *Objective Question* dimension and *Subjective Question* dimension, total 15 different tasks. Through a blend of objective and subjective tasks, the Advanced Knowledge Question Answering component challenges the models to demonstrate their insight in

areas ranging from chemical terminology and quantitative analysis to the recognition and interpretation of chemical structures and diagrams. The tasks within this section are designed to be both comprehensive and diagnostic, providing a clear measure of the models' readiness to tackle more advanced chemical inquiries.

### A.1.1 OBJECTIVE QUESTIONS (OBJQA)

The first dimension is objective question answering, which primarily assesses the model's grasp of fundamental chemical knowledge and its capability to apply this knowledge in straightforward scenarios. Objective question answering encompasses the following tasks: *Multiple Choice Task*, *Fill-in-the-Blank Task*, and *True/False Task*. By incorporating these tasks, ChemEval can more effectively gauge the model's overall proficiency in understanding and applying chemical knowledge across various contexts and formats.It should be noted that the *True/False Task* is exclusive to the text tasks and is not incorporated within the multimodal task set.

### A.1.2 SUBJECTIVE QUESTIONS (SUBJQA)

The second dimension is subjective question answering, which includes *Short Answer Task* and *Calculation Task*, both aiming to evaluate the depth of the model's comprehension and its ability to apply chemical knowledge effectively. Because on the basis of the previous task, the model also requires providing a detailed solution or reason, which involves the understanding of the chemical principles and concepts in the question, and applying these principles and concepts to construct logically clear and organized answers, which intuitively reflects the model's understanding of basic chemical knowledge.

Multimodal tasks further build upon these foundations, covering *Statistical Chart QA*, *Statistical Table QA* , *Reaction Profile Diagram QA*, *Theoretical Potential Energy Surface QA, Infrared Spectrum QA*, *Raman Spectrum QA*, *UV-Vis Spectrum QA*, *Diffraction Pattern QA* , *Kinetic Behavior Chart QA* and *Mass Spectrum QA*.These tasks comprehensively evaluate the model's ability to interpret and reason using chemical graphics and experimental data.

### A.2 LITERATURE UNDERSTANDING

Advanced Knowledge Question Answering is designed to assess the model's comprehension and mastery of chemical knowledge. In contrast, Literature Understanding evaluates the model's ability to interpret and assimilate information from chemical literature, which forms the foundation for downstream inductive generation tasks. Literature Understanding includes three dimensions: *Inductive Generation*, *Information Extraction*, and *Molecular Name Recognition*, comprising a total of 19 tasks. These tasks are crucial for understanding and extracting meaningful information from chemical literature. The primary focus is on assessing LLMs' ability to accurately extract and interpret chemical data from text, and to subsequently generate new, contextually relevant content. Importantly, such tasks are not covered by other chemical benchmarks. The following subsections detail the specific tasks.

### A.2.1 INFORMATION EXTRACTION (INFOE)

This is the first step to read a paper and also the foundation for the next inductive generation task. It involves the extraction of various elements related to chemistry, such as *named entities*, *reaction substrates*, and *catalyst types*, encompassing a total of 11 tasks. These tasks aim to decompose and organize chemical information found in text, covering entities, relationships, and various aspects of chemical reactions.

### A.2.2 INDUCTIVE GENERATION (INDUCGEN)

Based on Information Extraction, Inductive Generation involves creating new, coherent, and contextually relevant content based on existing data and knowledge. This process incorporates *Chemical Paper Abstract Generation*, *Research Outline Generation*, *Chemical Literature Topic Classification*, and *Reaction Type Recognition and Induction*, all focused on synthesizing and organizing chemical information in a coherent and meaningful manner.

### A.2.3 MOLECULAR NAME RECOGNITION(MNR)

Molecular Name Recognition is a foundational step in the extraction and organization of chemical information, focusing on the accurate identification of molecular names and related entities from scientific literature and data sources. This task goes beyond simple text extraction and leverages multimodal techniques to integrate information from textual, structural, and graphical data alike. Its subtasks encompass *Molecular Formula Recognition*, *Chemical Reaction Equation Recognition*, *2D Molecular Structure Recognition*, and *Synthetic Pathway Analysis*. Collectively, these subtasks enable comprehensive understanding and representation of chemical compounds and their transformations, serving as a crucial underpinning for downstream knowledge discovery and advanced reasoning in chemical informatics.

### A.3 MOLECULAR UNDERSTANDING

This section builds upon the previous foundation to assess the model's understanding and generative capabilities at the molecular level. It includes 4 dimensions: *Molecular Name Generation*, *Molecular Name Translation*, *Molecular Property Prediction*, and *Molecular Description*, a total of 15 tasks. Molecular Understanding explores tasks essential for molecular understanding, evaluating the LLMs' ability to generate, translate, and describe molecular names and properties. These tasks assess the models' proficiency in interpreting and generating chemical information accurately. The following subsections detail various specific tasks within this objective.

### A.3.1 MOLECULAR NAME GENERATION (MNGEN)

Molecular Name Generation is the basis of Molecular Understanding and only contains one task, *Molecular Name Generation from Text Description*. This task is purposed to evaluate the capacity of LLMs to generate valid chemical structure representations. It necessitates that the models, based on intricate textual descriptions encompassing molecular structures, properties, and classifications, synthesize SMILES molecular formulas effectively.

### A.3.2 MOLECULAR NAME TRANSLATION (MNTRANS)

Furthermore, Molecular Name Translation aims to enable a deep understanding of molecular structures and representations, which should serve as the fundamental knowledge for chemistry LLMs. It focuses on converting molecular names between different formats, requiring LLMs to output a specified alternative format based on a given molecular representation. It involves the conversion between representations of molecules such as *IUPAC names* and *SMILES* (Weininger, 1988) molecular formulas, encompassing a total of five tasks, each focusing on distinct aspects of molecular notation conversion.

### A.3.3 MOLECULAR PROPERTY PREDICTION (MPP)

Apart from molecular name understanding, the ability to predict molecular properties is also important. Molecular Property Prediction targets the forecast of a wide range of physical, chemical, and biological attributes of molecules, encapsulated in two core objectives: *Molecule Property Classification*, which predicts categories of properties such as ClinTox, HIV inhibition, and polarity; and *Molecule Property Regression*, focusing on estimating numerical values such as Lipophilicity, polarity, and boiling point.

### A.3.4 MOLECULAR DESCRIPTION (MOLDESC)

To facilitate a deeper assessment of molecular understanding, the Molecular Description task has been developed to comprehensively evaluate LLMs' capabilities in interpreting and describing molecular structures and their properties. This task consists of a series of subtasks, each requiring the prediction of physicochemical properties of molecules based on diverse input modalities. Besides the classic subtask of predicting physicochemical properties directly from molecular structures, this multimodal extension incorporates additional challenges: *Physicochemical Property Prediction from Infrared Spectrum*, *Physicochemical Property Prediction from Raman Spectrum*, *Physicochemical Property Prediction from UV-Vis Spectrum*, *Physicochemical Property Prediction from*

*Diffraction Pattern*, *Physicochemical Property Prediction from Mass Spectrum*, and *Physicochemical Property Prediction from NMR Spectrum*. Collectively, these tasks aim to assess LLMs' ability to interpret various molecular representations—spanning textual, graphical, and spectral data—for comprehensive property annotation and molecular understanding.

## A.4 SCIENTIFIC KNOWLEDGE DEDUCTION

Having established a solid grasp of basic chemical knowledge, the skill to interpret scientific literature, and the capacity to understand molecular structures, we expect that the model will proceed to conduct deeper chemical reasoning and deduction. So the part of Scientific Knowledge Deduction encompasses four key dimensions: *Retrosynthetic Analysis*, *Reaction Condition Recommendation*, *Reaction Outcome Prediction* and *Reaction Mechanism Analysis*, a total of 13 tasks, which are essential for effective chemical synthesis. This part evaluates the LLMs' capabilities in retrosynthetic analysis, recommending reaction conditions, predicting reaction outcomes, and analyzing reaction mechanisms. These tasks provide a comprehensive assessment of the models' performance in these critical areas of chemical synthesis.

### A.4.1 RETROSYNTHETIC ANALYSIS (RESSYN)

Retrosynthetic Analysis is a crucial technique in the field of chemical synthesis, particularly in organic synthesis. The process begins with the target product and then examines potential synthesis pathways and reactant substrates. This approach highlights the reverse reasoning capabilities of LLMs in the field of chemical synthesis. It comprises *Substrate Recommendation*, *Synthetic Pathway Recommendation* and *Synthetic Difficulty Evaluation*.

### A.4.2 REACTION CONDITION RECOMMENDATION (RCREC)

Based on the results of the Retrosynthetic Analysis, LLMs can recommend suitable reaction conditions. Reaction condition recommendation is a key task in chemical synthesis, involving selecting the most suitable conditions for specific chemical reactions to ensure maximum efficiency, selectivity, and yield. This task integrates recommendations for conditions such as *ligands*, *reagents*, and *catalysts*, encompassing a total of six tasks, each targeting a specific component of the reaction condition optimization.

### A.4.3 REACTION OUTCOME PREDICTION (ROP)

After determining the reaction pathway and reaction conditions, the large model can predict possible reaction outcomes. Reaction outcome prediction is a core technology in chemical synthesis aimed at predicting possible results of a reaction before it is actually carried out. This encompasses *Reaction Product Prediction*, *Product Yield Prediction*, *Reaction Rate Prediction*.

### A.4.4 REACTION MECHANISM ANALYSIS (RMA)

Reaction Mechanism Analysis is a critical area in the study of chemical reactions, aiming to explain the detailed steps involved in the transformation from reactants to products. This is the final step in the field of chemical synthesis, including identifying various intermediates, and transition states, as well as the kinetic and thermodynamic parameters of each step in the reaction. *Intermediate Derivation* is the sole subtask in this phase.

## B DETAILED DATASET STATISTICS

To comprehensively evaluate the chemical reasoning and knowledge capabilities of large language models, we constructed the ChemEval benchmark by integrating tasks from multiple sources. For the advanced knowledge question-answering level, our chemistry experts curated datasets based on textbooks and supplementary educational resources. For other levels, some tasks were adapted from existing datasets, such as ChemRxnExtractor (Guo et al., 2021), Mol-Instructions (Fang et al., 2023), ChemLLMBench (Guo et al., 2023), and SMolInstruct (Yu et al., 2024), while additional tasks were independently developed by our chemistry experts.

Table 4: Overview of text tasks in ChemEval.

| Tasks | Source | Number |
|---|---|---|
| Multiple Choice Task | Chemistry expert (Ours) | 50 |
| Fill-in-the-Blank Task | Chemistry expert (Ours) | 50 |
| True/False Task | Chemistry expert (Ours) | 50 |
| Short Answer Task | Chemistry expert (Ours) | 50 |
| Calculation Task | Chemistry expert (Ours) | 50 |
| Chemical Named Entity Recognition | Mol-Instructions | 50 |
| Chemical Entity Relationship Classification | Mol-Instructions | 50 |
| Synthetic Reaction Substrate Extraction | ChemRxnExtractor | 50 |
| Synthetic Reaction Additive Extraction | Chemistry expert (Ours) | 20 |
| Synthetic Reaction Solvent Extraction | Chemistry expert (Ours) | 20 |
| Reaction Temperature Extraction | Chemistry expert (Ours) | 20 |
| Reaction Time Extraction | Chemistry expert (Ours) | 20 |
| Reaction Product Extraction | ChemRxnExtractor | 50 |
| Characterization Method Extraction | Chemistry expert (Ours) | 20 |
| Catalysis Type Extraction | Chemistry expert (Ours) | 20 |
| Yield Extraction | Chemistry expert (Ours) | 20 |
| Chemical Paper Abstract Generation | Chemistry expert (Ours) | 20 |
| Research Outline Generation | Chemistry expert (Ours) | 20 |
| Chemical Literature Topic Classification | Chemistry expert (Ours) | 20 |
| Reaction Type Recognition and Induction | Chemistry expert (Ours) | 20 |
| Molecular Name Generation from Text Description | ChemLLMBench, Mol-Instructions, SMolInstruct | 50 |
| IUPAC to Molecular Formula | ChemLLMBench, SMolInstruct | 50 |
| SMILES to Molecular Formula | ChemLLMBench, SMolInstruct | 50 |
| IUPAC to SMILES Conversion | ChemLLMBench, SMolInstruct | 50 |
| SMILES to IUPAC Conversion | ChemLLMBench, SMolInstruct | 50 |
| SMILES to SELFIES and SELFIES to SMILES Translation | Chemistry expert (Ours) | 50 |
| Molecular Property Classification | Chemistry expert (Ours), ChemLLMBench, SMolInstruct | 260 |
| Molecular Property Regression | Chemistry expert (Ours), ChemLLMBench, Mol-Instructions, SMolInstruct | 220 |
| Physicochemical Property Prediction from Molecular Structure | ChemLLMBench, Mol-Instructions, SMolInstruct | 50 |
| Substrate Recommendation | ChemLLMBench, Mol-Instructions, SMolInstruct | 50 |
| Synthetic Pathway Recommendation | Chemistry expert (Ours) | 40 |
| Synthetic Difficulty Evaluation | Chemistry expert (Ours) | 20 |
| Ligand Recommendation | ChemLLMBench | 50 |
| Reagent Recommendation | ChemLLMBench | 50 |
| Solvent Recommendation | ChemLLMBench | 50 |
| Catalyst Recommendation | Chemistry expert (Ours) | 20 |
| Reaction Temperature Recommendation | Chemistry expert (Ours) | 20 |
| Reaction Time Recommendation | Chemistry expert (Ours) | 20 |
| Reaction Product Prediction | ChemLLMBench, Mol-Instructions, SMolInstruct | 50 |
| Product Yield Prediction | ChemLLMBench | 50 |
| Reaction Rate Prediction | Chemistry expert (Ours) | 20 |
| Intermediate Derivation | Chemistry expert (Ours) | 20 |
| **Total** | - | **1960** |

Tables 4 and 5 provide an overview of the sources and sizes of text and multimodal tasks included in ChemEval. The text subset contains 1,960 test examples, comprising 18 tasks sourced from other open datasets and 24 tasks designed in-house. The multimodal subset contains 1,200 test examples, integrating 12 tasks from other open datasets and 30 tasks independently designed by our team. Some tasks exist in both text and multimodal versions.

Table 6 summarizes the distribution of reaction types in our dataset. Coupling reactions account for the largest proportion, followed by substitution, oxidation/reduction, addition, elimination, and other reaction types. The "other" category covers 9 reaction types, including rearrangement, hydrolysis, cyclization, and so on. This dataset encompasses most major reaction types, providing comprehensive coverage for evaluating chemical reactions.

Overall, ChemEval integrates 25 tasks sourced from other open datasets and 37 custom-designed tasks developed in-house, with duplicates removed to prevent double counting. This design ensures the benchmark's diversity and comprehensiveness, allowing large language models to be evaluated across multiple dimensions of chemical knowledge and reasoning. By rigorously cross-checking against existing model training sets and leveraging previously unpublished laboratory data, we minimized potential risks of data leakage.

# C  DETAILED EXPERIMENTAL SETUPS

In this section, we introduce the details of our experimental setups, including the detailed description of the evaluated models and explanations of the metrics used in Section 3.2.5.

Table 5: Overview of multimodal tasks in ChemEval.

| Tasks | Source | Number |
|---|---|---|
| Multiple Choice Task | Chemistry expert (Ours) | 30 |
| Fill-in-the-Blank Task | Chemistry expert (Ours) | 20 |
| Statistical Chart QA | Chemistry expert (Ours) | 30 |
| Statistical Table QA | Chemistry expert (Ours) | 30 |
| Reaction Profile Diagram QA | Chemistry expert (Ours) | 20 |
| Theoretical Potential Energy Surface QA | Chemistry expert (Ours) | 20 |
| Infrared Spectrum QA | Chemistry expert (Ours) | 20 |
| Raman Spectrum QA | Chemistry expert (Ours) | 20 |
| UV-Vis Spectrum QA | Chemistry expert (Ours) | 20 |
| Diffraction Pattern QA | Chemistry expert (Ours) | 20 |
| Kinetic Behavior Chart QA | Chemistry expert (Ours) | 20 |
| Mass Spectrum QA | Chemistry expert (Ours) | 20 |
| Short Answer Task | Chemistry expert (Ours) | 20 |
| Calculation Task | Chemistry expert (Ours) | 30 |
| Molecular Formula Recognition | Chemistry expert (Ours) | 30 |
| Chemical Reaction Equation Recognition | Chemistry expert (Ours) | 30 |
| 2D Molecular Structure Recognition | Chemistry expert (Ours) | 60 |
| Synthetic Pathway Analysis | Chemistry expert (Ours) | 30 |
| IUPAC to Molecular Formula | ChemLLMBench, SMolInstruct | 20 |
| SMILES to Molecular Formula | ChemLLMBench, SMolInstruct | 20 |
| IUPAC to SMILES Conversion | ChemLLMBench, SMolInstruct | 20 |
| SMILES to IUPAC Conversion | ChemLLMBench, SMolInstruct | 50 |
| Molecular Property Classification | Chemistry expert (Ours), ChemLLMBench, SMolInstruct | 100 |
| Molecular Property Regression | Chemistry expert (Ours), ChemLLMBench, Mol-Instructions, SMolInstruct | 140 |
| Physicochemical Property Prediction from Infrared Spectrum | Chemistry expert (Ours) | 20 |
| Physicochemical Property Prediction from Raman Spectrum | Chemistry expert (Ours) | 20 |
| Physicochemical Property Prediction from UV-Vis Spectrum | Chemistry expert (Ours) | 20 |
| Physicochemical Property Prediction from Diffraction Pattern | Chemistry expert (Ours) | 20 |
| Physicochemical Property Prediction from Mass Spectrum | Chemistry expert (Ours) | 20 |
| Physicochemical Property Prediction from NMR Spectrum | Chemistry expert (Ours) | 20 |
| Substrate Recommendation | ChemLLMBench, Mol-Instructions, SMolInstruct | 20 |
| Synthetic Pathway Recommendation | Chemistry expert (Ours) | 40 |
| Synthetic Difficulty Evaluation | Chemistry expert (Ours) | 20 |
| Ligand Recommendation | ChemLLMBench | 20 |
| Reagent Recommendation | ChemLLMBench | 20 |
| Solvent Recommendation | ChemLLMBench | 20 |
| Catalyst Recommendation | Chemistry expert (Ours) | 20 |
| Reaction Temperature Recommendation | Chemistry expert (Ours) | 20 |
| Reaction Time Recommendation | Chemistry expert (Ours) | 20 |
| Reaction Product Prediction | ChemLLMBench, Mol-Instructions, SMolInstruct | 20 |
| Product Yield Prediction | ChemLLMBench | 20 |
| Intermediate Derivation | Chemistry expert (Ours) | 20 |
| **Total** | - | **1200** |

## C.1 MODELS

In order to comprehensively assess the scientific capabilities of Large Language Models (LLMs), we evaluate several high-performing LLMs that are widely accessible, including general and specialized models. These models are selected to represent a diverse range of organizations and vary in size.

- **GPT-4o**: GPT-4o is OpenAI's latest flagship multimodal large language model, capable of processing and generating text, audio, and images through a unified architecture for seamless cross-modal reasoning and interaction. It sets new benchmarks in multilingual, speech, and visual understanding, exhibiting advanced performance with significantly improved speed and efficiency compared to previous models.

- **OpenAI-o1/o3-mini**: OpenAI o1 and o3-mini are lightweight, cost-effective reasoning models that deliver strong performance in science, mathematics, and programming tasks while offering significantly improved response speed and reliability compared to their predecessors, making them well-suited for rapid, real-world applications.

Table 6: Distribution of reaction types in ChemEval

| Reaction Type | Counts | Percentage |
|---|---|---|
| Coupling reactions | 321 | 62.4% |
| Substitution reactions | 81 | 15.8% |
| Oxidation/reduction reactions | 51 | 9.9% |
| Addition reactions | 24 | 4.7% |
| Elimination reactions | 13 | 2.5% |
| Other reactions | 24 | 4.7% |

Table 7: Performance overview of multi-level 0-shot text tasks on ChemEval (part 1). Claude3.7T denotes Claude 3.7-Sonnet-Thinking, whereas Claude3.7N denotes Claude 3.7-Sonnet.

| Dimension | Task | Metric | OpenAI-o3-mini | OpenAI-o1 | GPT-4o | Claude3.7T | Claude3.7N | Deepseek-R1 | Deepseek-V3 | Qwen2.5-72B | Qwen2.5-32B |
|---|---|---|---|---|---|---|---|---|---|---|---|
| *Advanced Knowledge Question Answering* | | | | | | | | | | | |
| ObjQA | MCTask | Accuracy | 72.00 | 74.00 | 66.80 | 62.80 | 60.80 | 82.40 | 76.00 | 67.20 | 67.20 |
| ObjQA | FBTask | LLM Score | 62.42 | 60.92 | 51.19 | 45.28 | 44.73 | 59.41 | 63.88 | 53.92 | 50.93 |
| ObjQA | TFTask | Accuracy | 68.00 | 46.00 | 57.60 | 58.80 | 58.00 | 75.20 | 67.20 | 58.40 | 49.20 |
| SubjQA | SATask | LLM Score | 68.00 | 64.50 | 61.20 | 56.70 | 55.10 | 68.50 | 71.70 | 58.50 | 58.10 |
| SubjQA | CalcTask | LLM Score | 75.50 | 78.00 | 61.80 | 55.74 | 53.60 | 76.10 | 79.20 | 61.90 | 57.40 |
| *Literature Understanding* | | | | | | | | | | | |
| InfoE | CNER | F1 | 61.30 | 64.56 | 65.76 | 60.21 | 54.55 | 64.14 | 60.85 | 61.61 | 56.33 |
| InfoE | CERC | F1 | 29.65 | 22.37 | 25.66 | 25.19 | 24.77 | 27.18 | 24.94 | 26.05 | 27.21 |
| InfoE | SubE | Accuracy | 66.91 | **73.71** | 66.32 | 61.59 | 65.76 | 75.18 | 61.26 | 62.56 | 58.05 |
| InfoE | AddE | F1 | 76.67 | 81.67 | **85.00** | 79.33 | 81.10 | 82.67 | 80.67 | 84.00 | 80 |
| InfoE | SolvE | F1 | 89.00 | 86.50 | 85.00 | 87.60 | 84.30 | **90.20** | 88.50 | 85.00 | 90.00 |
| InfoE | TempE | F1 | 65.00 | 70.00 | 67.00 | 72.00 | 69.00 | 65.00 | 72.00 | 65.00 | 62.00 |
| InfoE | TimeE | F1 | 95.00 | 95.00 | 95.00 | 95.00 | 95.00 | 95.00 | 95.00 | 90.00 | 95.00 |
| InfoE | ProdE | Accuracy | 87.62 | 90.25 | 86.09 | 82.39 | 85.04 | 91.20 | 87.52 | 84.86 | 76.38 |
| InfoE | CharME | F1 | 66.67 | 51.67 | 72.85 | 81.01 | 71.84 | 21.33 | **81.80** | 74.57 | 79.42 |
| InfoE | CatTE | F1 | 65.00 | 95.00 | 94.00 | 82.00 | 77.00 | 99.00 | **100.00** | **100.00** | **100.00** |
| InfoE | YieldE | F1 | 65.00 | **85.00** | 79.00 | 61.00 | 59.00 | 77.70 | 65.00 | 65.00 | 78.00 |
| InducGen | AbsGen | LLM Score | 68.75 | 63.75 | 63.00 | 63.00 | 66.75 | 65.00 | 64.75 | 64.75 | 60.00 |
| InducGen | OLGen | LLM Score | 35.00 | 25.00 | 35.50 | 26.50 | 28.50 | 37.00 | 27.00 | 24.25 | 29.75 |
| InducGen | TopC | Accuracy | 50.00 | 55.00 | 49.00 | 56.00 | 51.00 | 57.00 | 50.00 | 64.00 | 35.00 |
| InducGen | ReactTR | F1 | 20.00 | 25.00 | **32.00** | 29.00 | 26.00 | 21.00 | 28.00 | 22.00 | 26.00 |
| *Molecular Understanding* | | | | | | | | | | | |
| MNGen | MolNG | Tanimoto (valid) | 51.58 (78%) | 49.80 (72%) | 39.30 (89%) | 33.85 (70%) | 42.28 (78%) | 56.05 (87%) | 51.19 (96%) | 20.58 (79%) | 14.60 (64%) |
| MNTrans | IUPAC2MF | L2 | 0.6214 | 0.7737 | 0.5304 | 0.3252 | 0.3349 | 0.6026 | 0.6176 | 0.3407 | 0.3070 |
| MNTrans | SMILES2MF | L2 | 0.6276 | 0.6330 | 0.3627 | 0.3618 | 0.3468 | 0.4402 | 0.3563 | 0.2448 | 0.2548 |
| MNTrans | IUPAC2SMILES | Tanimoto (valid) | 29.61 (42%) | 29.72 (50%) | 34.71 (83%) | 31.89 (68%) | 39.12 (72%) | 30.70 (63%) | 46.07 (88%) | 15.90 (76%) | 10.55 (59%) |
| MNTrans | SMILES2IUPAC | Exact Match | 0.00 | 0.00 | 0.00 | 0.00 | 0.00 | 1.20 | 0.00 | 0.00 | 0.00 |
| MNTrans | SMILES2IUPAC | BLEU | 4.37 | 3.24 | 0.96 | 3.27 | 3.46 | 4.17 | 1.67 | 0.33 | 0.15 |
| MNTrans | SMILES2IUPAC | Tanimoto | 0.00 | 0.00 | 12.08 | 22.73 | 24.99 | 25.90 | 19.16 | 13.01 | 8.68 |
| MNTrans | S2S | Tanimoto (valid) | 9.76 (30%) | 9.72 (42%) | 13.41 (62%) | 9.37 (40%) | 10.58 (44%) | 16.04 (71%) | 16.27 (62%) | 11.47 (50%) | 6.93 (37%) |
| MPP | MolPC | Accuracy | 72.88 | 67.50 | 64.57 | 58.90 | 54.37 | 53.54 | 48.73 | 48.13 | 67.70 |
| MPP | MolPR | NRMSE (valid) | 12.7593 (99%) | 12.3852 (99%) | 9.9322 (51%) | 13.9702 (92%) | 14.0966 (96%) | 15.8881 (100%) | 8.3675 (98%) | 13.0756 (100%) | 17.6710 (91%) |
| MolDesc | Mol2PC | LLM Score | 19.50 | 19.00 | 7.00 | 8.00 | 8.00 | 11.90 | 13.50 | 20.80 | 5.90 |
| *Scientific Knowledge Deduction* | | | | | | | | | | | |
| ReSyn | SubRec | F1 | 4.67 | 1.00 | 0.00 | 1.46 | 1.77 | 1.63 | 2.27 | 1.06 | 0.20 |
| ReSyn | PathRec | LLM Score | 49.38 | 30.63 | 22.88 | 0.36 | 41.88 | 52.75 | 37.38 | 41.13 | 36.88 |
| ReSyn | SynDE | NRMSE (valid) | 5.4045 (20%) | - (5%) | - (0%) | - (0%) | 1.9854 (39%) | - (0%) | - (0%) | 0.2670 (100%) | - (0%) |
| RCRec | LRec | F1 | 4.00 | 0.00 | 13.20 | 2.00 | 4.40 | 6.80 | 7.60 | 4.40 | 8.00 |
| RCRec | RRec | F1 | 32.00 | 25.64 | 15.80 | 27.43 | 25.80 | 21.93 | 8.35 | 37.75 | 34.56 |
| RCRec | SolvRec | F1 | 16.00 | 10.00 | 20.40 | 18.80 | 17.60 | 22.40 | 24.00 | 50.40 | 51.60 |
| RCRec | CatRec | F1 | 0.00 | 0.00 | 0.00 | 0.00 | 0.00 | 0.00 | 0.00 | 0.00 | 0.00 |
| RCRec | TempRec | NRMSE (valid) | 0.2201 (100%) | 0.3278 (100%) | 0.2545 (100%) | 0.2263 (100%) | 0.5398 (100%) | 0.2078 (100%) | 0.2096 (100%) | 0.3782 (100%) | 0.2475 (100%) |
| RCRec | TimeRec | NRMSE (valid) | 0.2165 (100%) | 0.2746 (100%) | 0.2468 (100%) | 0.3662 (100%) | 0.4008 (100%) | 0.2291 (100%) | 0.2579 (100%) | **0.2022 (100%)** | 0.2377 (100%) |
| ROP | PPred | F1 | 10.00 | 21.33 | 1.67 | 12.27 | 16.16 | 11.97 | 0.93 | 1.73 | 0.53 |
| ROP | YPred | Accuracy | 8.00 | 12.00 | 43.50 | 16.00 | 9.00 | 11.00 | 22.50 | 26.00 | **85.00** |
| ROP | RatePred | Overlap | 16.74 | 21.08 | 13.81 | 9.06 | 7.21 | 17.12 | 17.71 | 10.71 | 9.48 |
| RMA | IMDer | LLM Score | 80.00 | 80.00 | 81.50 | 81.50 | 81.00 | 79.50 | 80.50 | 77.25 | 79.00 |

- **Claude-3.7-Sonnet**: Claude 3.7 Sonnet is Anthropic's most advanced hybrid reasoning language model to date, integrating rapid response with deep, stepwise analytical capabilities and offering flexible dual modes for both instant answers and complex multi-stage problem-solving across a range of scientific and coding tasks.

- **Gemini-2.5-pro**: Gemini 2.5 Pro is Google DeepMind's latest multimodal large language model that integrates advanced "thinking" mechanisms and hybrid attention architectures, enabling state-of-the-art reasoning, code generation, and long-context understanding across text, image, audio, and video inputs, with support for up to one million tokens in a single context window.

- **Grok3**: Grok 3 is a new generation of large language model developed by xAI. It has achieved breakthroughs in key benchmark tests such as mathematical reasoning, scientific logical reasoning, and code writing. In addition, it supports multimodal interaction and can also access real-time information through the X platform to enhance the timeliness and accuracy of its responses.

- **DeepSeek-V3**: DeepSeek-V3 is a powerful 671-billion-parameter Mixture-of-Experts (MoE) language model developed by DeepSeek, trained on 14.8 trillion tokens with innovations like Multi-head Latent Attention (MLA) and Multi-Token Prediction (MTP) to achieve state-of-the-art performance in mathematics, coding, and multilingual tasks. It features a 128K context window and efficient inference, with future versions expected to include multi-modal capabilities.

- **DeepSeek-R1**: DeepSeek-R1 is a reasoning-optimized model based on the DeepSeek-V3-Base architecture. It is trained with reinforcement learning and human feedback to enhance its performance in complex reasoning tasks such as logical deduction and mathematical problem-solving while maintaining high safety and reliability.

- **Qwen2.5-7B/14B/32B/72B**: Qwen 2.5 is a series of advanced large language models developed by Alibaba Cloud, featuring models with parameter sizes ranging from 0.5B to 72B. These models have significantly improved capabilities in areas such as coding, mathematics, and multilingual support, and they are trained on a large-scale dataset of up to 18 trillion tokens

- **LLaMA3.3-8B** : Meta Llama 3 8B is a powerful large language model with 8 billion parameters, optimized for dialogue and text generation. It is trained on over 15 trillion tokens and features a 128K token vocabulary and Grouped-Query Attention for enhanced performance.

Table 8: Performance overview of multi-level 0-shot text tasks on ChemEval (Part 2).

| Dimension | Task | Metric | Qwen2.5-14B | Qwen2.5-7B | Llama3.3-8B | Grok3 | Gemini-2.5-Pro | ChemDFM | ChemLLM | LlaSMol | ChemSpark | Chemcrow |
|---|---|---|---|---|---|---|---|---|---|---|---|---|
| *Advanced Knowledge Question Answering* | | | | | | | | | | | | |
| ObjQA | MCTask | Accuracy | 64.80 | 59.60 | 40.40 | 68.80 | **87.60** | 41.20 | 24.40 | 24.00 | 43.60 | 58.00 |
| ObjQA | FBTask | LLM Score | 45.76 | 39.52 | 34.17 | 54.36 | **63.95** | 24.16 | 34.97 | 13.92 | 24.57 | 43.14 |
| ObjQA | TFTask | Accuracy | 52.00 | 55.20 | 46.00 | 64.40 | **77.60** | 46.00 | 19.20 | 58.00 | 50.00 | 74.00 |
| SubjQA | SATask | LLM Score | 57.20 | 50.80 | 38.40 | **73.59** | 72.00 | 32.20 | 13.20 | 14.50 | 33.60 | 43.50 |
| SubjQA | CalcTask | LLM Score | 50.80 | 43.60 | 28.00 | 81.20 | **82.40** | 14.70 | 15.90 | 7.50 | 18.50 | 43.50 |
| *Literature Understanding* | | | | | | | | | | | | |
| InfoE | CNER | F1 | 46.31 | 61.27 | 55.34 | 60.75 | 68.30 | 41.17 | 0.16 | 11.62 | **71.44** | 57.46 |
| InfoE | CERC | F1 | 28.19 | 26.10 | 17.31 | 26.04 | 25.43 | 8.74 | 0.24 | 1.24 | **39.27** | 22.05 |
| InfoE | SubE | Accuracy | 59.61 | 58.43 | 64.02 | 72.87 | 72.05 | 20.07 | 0.00 | 0.00 | 74.38 | 50.91 |
| InfoE | AddE | F1 | 83.00 | 61.67 | 45.81 | 85.00 | 95.00 | 45.00 | 0.00 | 0.00 | 65.00 | 43.33 |
| InfoE | SolvE | F1 | 86.50 | 82.50 | 75.47 | 85.00 | 83.17 | 80.50 | 1.67 | 0.00 | 83.79 | 87.50 |
| InfoE | TempE | F1 | 70.00 | 65.00 | 62.00 | 70.00 | 69.00 | 74.33 | 3.23 | 0.00 | **83.00** | 65.00 |
| InfoE | TimeE | F1 | **95.00** | **95.00** | 90.00 | **95.00** | 94.00 | 78.00 | 23.10 | 25.00 | **95.00** | **95.00** |
| InfoE | ProdE | Accuracy | 82.44 | 77.00 | 74.54 | 91.04 | 92.82 | 34.73 | 0.00 | 0.00 | **94.40** | 71.38 |
| InfoE | CharME | F1 | 67.92 | 43.00 | 44.18 | 79.36 | 73.11 | 27.26 | 0.00 | 0.00 | 12.98 | 25.00 |
| InfoE | CatTE | F1 | 75.00 | 64.00 | 65.00 | 97.00 | 96.00 | 49.00 | 0.00 | 5.00 | 31.00 | 85.00 |
| InfoE | YieldE | F1 | 80.00 | 67.00 | 46.00 | 61.00 | 74.00 | 45.00 | 0.00 | 5.00 | 61.00 | 50.00 |
| InducGen | AbsGen | LLM Score | 59.25 | 54.75 | 62.00 | 69.50 | **67.25** | 0.00 | 5.50 | 26.25 | 38.25 | 57.50 |
| InducGen | OLGen | LLM Score | 29.75 | 27.75 | 22.75 | 35.25 | **39.50** | 0.00 | 3.75 | 31.25 | 30.50 | 32.50 |
| InducGen | TopC | Accuracy | 45.00 | 41.00 | 32.00 | 47.00 | **67.00** | 51.00 | 0.00 | 0.00 | 30.00 | 45.00 |
| InducGen | ReactTR | F1 | 26.00 | 31.00 | 26.00 | 28.00 | 31.00 | 13.00 | 0.00 | 5.00 | 17.00 | 5.00 |
| *Molecular Understanding* | | | | | | | | | | | | |
| MNGen | MolNG | Tanimoto (valid) | 11.03 (53%) | 3.92 (32%) | 5.83 (40%) | 57.86 (94%) | **71.11 (93%)** | 47.06 (69%) | 0.00 (0%) | 3.71 (76%) | 74.81 (98%) | 40.92 (90%) |
| MNTrans | IUPAC2MF | L2 | 0.3126 | 0.1856 | 0.2433 | 0.7110 | **0.8382** | 0.6119 | 0.0454 | 0.0000 | 0.8807 | 0.1408 |
| MNTrans | SMILES2MF | L2 | 0.2114 | 0.0980 | 0.1728 | 0.3980 | **0.6574** | 0.6399 | 0.0375 | 0.0000 | 0.8133 | 0.3089 |
| MNTrans | IUPAC2SMILES | Tanimoto (valid) | 8.18 (52%) | 3.46 (30%) | 5.24 (30%) | 65.81 (94%) | 61.35 (87%) | 46.71 (88%) | 0.00 (100%) | 4.70 (56%) | **87.84 (100%)** | 25.68 (64%) |
| MNTrans | SMILES2IUPAC | Exact Match | 0.00 | 0.00 | 0.00 | 1.20 | 1.20 | 0.00 | 0.00 | 0.00 | **14.00** | 0.00 |
| MNTrans | SMILES2IUPAC | BLEU | 0.22 | 0.00 | 0.44 | 4.69 | 13.55 | 0.56 | 0.00 | 0.00 | **48.25** | 0.38 |
| MNTrans | SMILES2IUPAC | Tanimoto | 5.76 | 3.78 | 3.71 | 30.47 | 56.82 | 2.06 | 0.00 | 2.22 | **66.26** | 0.00 |
| MNTrans | S2S | Tanimoto (valid) | 10.52 (60%) | 2.28 (14%) | 1.74 (12%) | 17.56 (59%) | 13.13 (44%) | 2.12 (25%) | 0.00 (50%) | 0.60 (48%) | **87.36 (94%)** | 9.83 (38%) |
| MPP | MolPC | Accuracy | 64.22 | 64.05 | 47.26 | 56.61 | 63.63 | 61.35 | 0.00 | 46.50 | **85.57** | 46.00 |
| MPP | MolPR | NRMSE (valid) | 11.7005 (90%) | 8.5890 (98%) | 61.4736 (62%) | 9.0283 (100%) | 11.7270 (100%) | 394.9424 (83%) | 179.3606 (93%) | 29.9686 (73%) | 1.2142 (100%) | **0.3408 (38%)** |
| MolDesc | Mol2PC | LLM Score | 7.20 | 14.50 | 2.10 | 28.00 | 0.70 | 3.10 | 0.30 | 0.00 | **48.90** | 21.00 |
| *Scientific Knowledge Deduction* | | | | | | | | | | | | |
| ReSyn | SubRec | F1 | 0.00 | 1.42 | 0.27 | 0.87 | 0.00 | 3.99 | 0.00 | 0.00 | **12.37** | 0.00 |
| ReSyn | PathRec | LLM Score | 32.63 | 27.13 | 20.88 | 32.13 | 43.75 | 24.13 | 10.88 | 10.00 | 38.75 | 48.75 |
| ReSyn | SynDE | NRMSE (valid) | 0.3551 (100%) | - (0%) | - (0%) | - (0%) | - (0%) | - (0%) | 33.0049 (78%) | 1.2374 (45%) | 1.7992 (87%) | - (0%) |
| RCRec | LRec | F1 | 6.80 | 2.80 | 2.13 | 36.00 | 0.00 | 26.00 | 0.00 | 0.00 | **37.60** | 18.00 |
| RCRec | RRec | F1 | 37.65 | 16.93 | 8.78 | 44.60 | 0.73 | 13.13 | 0.00 | 0.50 | **63.72** | 36.65 |
| RCRec | SolvRec | F1 | 15.60 | 25.60 | 3.63 | 24.00 | 0.00 | 10.53 | 0.00 | 0.50 | **30.40** | 12.00 |
| RCRec | CatRec | F1 | 0.00 | 0.00 | 0.00 | 0.00 | 0.00 | 0.00 | 0.00 | 0.00 | **0.50** | 0.00 |
| RCRec | TempRec | NRMSE (valid) | 0.1989 (100%) | 0.3223 (100%) | - (0%) | 0.1972 (100%) | **0.1814 (100%)** | 0.3811 (99%) | 1.1184 (98%) | 0.8658 (100%) | 0.2742 (100%) | 0.2392 (100%) |
| RCRec | TimeRec | NRMSE (valid) | 0.2505 (100%) | 0.3213 (100%) | - (0%) | 0.2164 (100%) | 0.2425 (100%) | 0.4732 (100%) | 1.7937 (98%) | 0.4351 (80%) | 0.3937 (100%) | 0.5209 (70%) |
| ROP | PPred | F1 | 0.00 | 0.00 | 0.00 | 11.33 | 29.20 | 18.80 | 0.00 | 16.00 | **56.40** | 0.00 |
| ROP | YPred | Accuracy | 33.50 | 67.00 | 35.50 | 8.00 | 17.50 | 7.20 | 0.00 | 28.00 | 72.00 | 24.00 |
| ROP | RatePred | Overlap | 9.54 | 13.35 | 6.92 | 8.77 | **27.01** | 3.79 | 0.00 | 3.68 | 2.90 | 0.00 |
| RMA | IMDer | LLM Score | 67.75 | 78.75 | 81.25 | 81.25 | 82.25 | 76.00 | 4.75 | 1.50 | **92.75** | 28.75 |

- **Qwen-VL Max**: Qwen-VL-Max is the most capable large visual language model in the Qwen-VL series, offering optimal performance on a broad range of complex tasks. It has significantly enhanced visual reasoning and instruction-following abilities, and can handle high-definition images with resolutions above one million pixels.

- **Phi-Vision-3.5**: Phi-3.5-vision is a lightweight, state-of-the-art open multimodal model developed by Microsoft, with 4.2B parameters and a 128K context length. It excels in handling both text and visual inputs, offering capabilities in general image understanding, optical character recognition, chart interpretation, and video summarization.

- **ChemDFM**: ChemDFM is a pioneering large language model (LLM) specifically designed for chemistry, trained on 34 billion tokens from chemical literature and textbooks and fine-tuned using 2.7 million instructions. It demonstrates superior performance in various chemical tasks such as molecule recognition, molecular property prediction, and reaction analysis, significantly outperforming most representative open-source LLMs.

- **LlaSMol**: LlaSMol is a series of large language models fine-tuned on a large-scale, comprehensive, and high-quality instruction tuning dataset named SMolInstruct for chemistry tasks. These models, based on open-source LLMs like Galactica, Llama 2, Code Llama, and Mistral, demonstrate strong performance on various chemistry tasks, significantly outperforming previous LLMs and approaching the performance of state-of-the-art task-specific models. We select the Mistral-based version for experiments due to its superior performance.

- **ChemLLM**: ChemLLM is the first specialized large language model dedicated to chemistry, trained on a unique dataset ChemData, and evaluated on a comprehensive benchmark ChemBench. This model shows remarkable capabilities in handling various chemistry tasks and exhibits strong general language skills.

- **ChemSpark**: ChemSpark, formally known as Spark-Chemistry-X1-13B, is a LLM specialized for chemistry developed by iFLYTEK and released on the ModelScope platform. It was created by fine-tuning the iFLYTEK Spark-X1 base model on various chemical task datasets.

Table 9: Performance overview of multimodal tasks on ChemEval.

| Dimension | Task | Metric | GLM-4V | GPT-4o | Claude3.7T | Qwen-vl-max | Phi-vision-3.5 | Gemini-2.5-Pro |
|---|---|---|---|---|---|---|---|---|
| | | | *Advanced Knowledge Question Answering* | | | | | |
| ObjQA | MCTask | Accuracy | 32.22 | 40.86 | 7.78 | 43.33 | 35.56 | **45.55** |
| ObjQA | FBTask | Accuracy | 36.67 | 52.41 | 17.77 | 48.12 | 15.02 | **58.80** |
| SubjQA | SCQA | LLM Score | 65.33 | 68.67 | 30.22 | **82.00** | 44.44 | 80.89 |
| SubjQA | STQA | LLM Score | 64.22 | 54.22 | 32.67 | 72.22 | 32.67 | **76.22** |
| SubjQA | RPDQA | LLM Score | 50.67 | 62.93 | 20.00 | **70.67** | 37.67 | 70.00 |
| SubjQA | TPESQA | LLM Score | 62.33 | 69.33 | 21.67 | **76.33** | 45.67 | 70.67 |
| SubjQA | IRSQA | LLM Score | 53.33 | 59.00 | 35.33 | 62.33 | 42.00 | **66.33** |
| SubjQA | RSQA | LLM Score | 64.33 | 70.00 | 35.67 | 71.33 | 51.33 | **76.00** |
| SubjQA | UVSQA | LLM Score | 62.67 | 62.67 | 33.33 | 66.00 | 48.00 | **69.33** |
| SubjQA | DPQA | LLM Score | 67.00 | 75.67 | 37.00 | 83.33 | 51.00 | **76.00** |
| SubjQA | KBCQA | LLM Score | 68.33 | 77.00 | 48.67 | **81.67** | 51.00 | 79.33 |
| SubjQA | MSQA | LLM Score | 66.33 | 74.40 | 22.00 | **83.67** | 46.33 | 72.00 |
| SubjQA | SATask | LLM Score | 46.67 | 55.28 | 46.33 | 57.67 | 35.00 | **71.00** |
| SubjQA | CalcTask | LLM Score | 49.11 | 60.67 | 51.78 | 62.00 | 36.89 | **79.78** |
| | | | *Literature Understanding* | | | | | |
| MNR | MFR | Accuracy | **100.00** | 95.56 | 2.22 | **100.00** | 85.55 | 84.45 |
| MNR | CRER | Accuracy | **95.56** | 93.34 | 3.33 | 93.33 | 15.56 | 42.22 |
| MNR | 2DMolR | Tanimoto | 3.73 | **20.92** | 0.00 | 16.26 | 1.98 | - |
| MNR | PathA | F1 | 0.00 | 0.00 | 0.00 | 0.00 | 0.00 | - |
| | | | *Molecular Understanding* | | | | | |
| MNTrans | IUPAC2MF | L2 | 0.3048 | 0.5653 | 0.2106 | 0.1175 | 0.1690 | **0.5892** |
| MNTrans | SMILES2MF | L2 | 0.1251 | 0.2144 | 0.0468 | 0.1367 | 0.1018 | **0.4951** |
| MNTrans | IUPAC2SMILES | Tanimoto | 8.40 | 44.43 | 11.90 | 24.63 | 4.37 | **77.19** |
| MNTrans | SMILES2IUPAC | Exact | 0.00 | 0.00 | 0.00 | 0.00 | 0.00 | **2.00** |
| MNTrans | SMILES2IUPAC | BLEU | 23.15 | 19.04 | 22.81 | 24.44 | **26.19** | 18.47 |
| MNTrans | SMILES2IUPAC | Tanimoto | 1.73 | 2.09 | **8.88** | 0.74 | 1.22 | 4.16 |
| MPP | MolPC | Accuracy | 50.51 | 49.70 | 54.67 | 58.32 | 53.75 | **62.08** |
| MPP | MolPR | NRMSE (valid) | 2.3782 (57%) | 1.0268 (71%) | **0.3491** (29%) | 21.8799 (100%) | 3.0580 (43%) | 16.1085 (100%) |
| MolDesc | IRS2PC | LLM Score | 54.00 | 58.00 | **66.33** | 60.67 | 45.00 | 60.67 |
| MolDesc | RS2PC | LLM Score | 44.00 | 51.67 | **63.00** | 57.67 | 38.33 | 55.33 |
| MolDesc | UV2PC | LLM Score | 54.67 | 59.67 | 65.67 | 63.00 | 40.67 | **67.00** |
| MolDesc | DP2PC | LLM Score | 58.33 | 65.00 | **74.00** | 69.00 | 41.33 | 69.33 |
| MolDesc | MS2PC | LLM Score | 54.33 | 61.67 | **75.33** | 67.00 | 38.67 | 69.00 |
| MolDesc | NMR2PC | LLM Score | 54.33 | 65.00 | **71.67** | 68.33 | 37.67 | 66.67 |
| | | | *Scientific Knowledge Deduction* | | | | | |
| ReSyn | SubRec | F1 | 0.00 | 0.00 | 0.00 | **1.48** | 0.00 | **1.48** |
| ReSyn | PathRec | LLM Score | 45.00 | 57.00 | **67.00** | 54.67 | 31.67 | 61.67 |
| ReSyn | SynDE | NRMSE | 0.4220 | 0.3199 | 0.5575 | **0.2234** | - | 0.5437 |
| RCRec | LRec | F1 | 0.00 | **28.33** | 1.67 | 8.33 | 11.67 | 5.00 |
| RCRec | RRec | F1 | 0.00 | 5.00 | 5.00 | 6.67 | 6.67 | **8.33** |
| RCRec | SolvRec | F1 | 15.00 | 23.33 | 21.67 | **30.00** | 18.33 | 28.33 |
| RCRec | CatRec | F1 | 0.00 | 0.00 | 0.00 | 0.00 | 0.00 | 0.00 |
| RCRec | TempRec | NRMSE | **0.1220** | 0.4845 | 0.3913 | 0.5346 | - | 0.1777 |
| RCRec | TimeRec | NRMSE | - | - | **0.4378** | - | - | - |
| ROP | PRec | F1 | 0.00 | 0.00 | 0.00 | **3.33** | 0.00 | 1.67 |
| ROP | YPred | Accuracy | - | 43.33 | 20.00 | 25.00 | **78.33** | 31.67 |
| RMA | IMPred | LLM Score | 67.67 | 71.33 | 76.67 | 62.33 | 35.00 | **77.67** |

## C.2 METRICS

In this study, we employ a variety of evaluation metrics to fine-grained assess model performance across different tasks. The main metrics include:

- **F1 Score and Accuracy:** These are the primary metrics used for most tasks. The F1 score combines precision and recall to evaluate classification performance, while accuracy measures the proportion of correct predictions.

- **BLEU:** Calculated by comparing the n-gram overlap between the model-generated text and the reference answer, incorporating a brevity penalty to penalize overly short outputs. This metric is mainly used to assess the similarity between generated results and reference answers.

- **Exact Match:** This metric checks whether the model output exactly matches the ground truth.

- **Normalized Root Mean Square Error (NRMSE):** Used to evaluate the prediction error in numerical or regression tasks, and lower values indicate better model performance.

- **Valid Output Ratio:** The proportion of valid outputs provided by the model.

- **LLM Score:** Subjective evaluation by other large language models, focusing on the reasonableness and completeness of the answers.

- **L2 Score (L2):** An indicator for evaluating the similarity between molecular formulas. Specifically, L2 Score is calculated as $1/(1 + \text{L2 distance})$, where the L2 distance refers to the L2 norm

Table 10: Performance overview of multi-level 3-shot text tasks on ChemEval (part 1). Claude3.7T denotes Claude 3.7-Sonnet-Thinking, whereas Claude3.7N denotes Claude 3.7-Sonnet.

| Dimension | Task | Metric | OpenAI-o3-mini | OpenAI-o1 | GPT-4o | Claude3.7T | Claude3.7N | Deepseek-R1 | Deepseek-V3 | Qwen2.5-72B | Qwen2.5-32B |
|---|---|---|---|---|---|---|---|---|---|---|---|
| *Advanced Knowledge Question Answering* | | | | | | | | | | | |
| ObjQA | MCTask | Accuracy | 72.00 | 82.00 | 69.20 | 65.20 | 65.20 | 82.40 | 72.00 | 68.00 | 71.20 |
| ObjQA | FBTask | LLM Score | 51.46 | **62.65** | 45.59 | 42.56 | 42.28 | 59.96 | 57.89 | 53.53 | 45.99 |
| ObjQA | TFTask | Accuracy | 76.00 | **86.00** | 66.00 | 57.60 | 62.40 | 80.80 | 72.80 | 48.40 | 59.60 |
| SubjQA | SATask | LLM Score | 67.00 | 68.50 | 61.00 | 54.10 | 53.90 | 71.40 | 70.40 | 60.80 | 55.90 |
| SubjQA | CalcTask | LLM Score | 75.00 | 78.50 | 59.10 | 53.73 | 55.40 | 75.10 | 77.40 | 61.61 | 52.61 |
| *Literature Understanding* | | | | | | | | | | | |
| InfoE | CNER | F1 | 66.33 | 70.59 | 71.14 | 64.62 | 62.18 | 70.85 | 63.28 | 65.92 | 59.45 |
| InfoE | CERC | F1 | 29.30 | 32.69 | 25.72 | 23.11 | 25.39 | 29.11 | 25.65 | 25.63 | 26.18 |
| InfoE | SubE | Accuracy | 73.17 | **78.01** | 65.93 | 62.66 | 61.55 | 76.88 | 75.78 | 70.10 | 60.62 |
| InfoE | AddE | F1 | 88.33 | **95.67** | 90.94 | 90.57 | 92.63 | 89.57 | 90.87 | 88.80 | 81.84 |
| InfoE | SolvE | F1 | 84.00 | 85.00 | 80.00 | 81.50 | 84.63 | 85.00 | 81.60 | 75.00 | 84.00 |
| InfoE | TempE | F1 | 70.00 | 75.00 | 73.00 | 80.00 | 80.00 | **83.00** | 80.00 | 80.00 | 75.00 |
| InfoE | TimeE | F1 | 95.00 | 95.00 | 95.00 | 95.00 | 95.00 | 95.00 | 95.00 | 95.00 | 95.00 |
| InfoE | ProdE | Accuracy | 88.06 | 91.48 | 86.88 | 82.35 | 87.34 | 92.33 | 91.75 | 84.05 | 71.38 |
| InfoE | CharME | F1 | 76.02 | 79.60 | 78.97 | 77.88 | 75.02 | 77.86 | 77.34 | 73.63 | 72.18 |
| InfoE | CatTE | F1 | 95.00 | 95.00 | 98.00 | 91.00 | 94.00 | **100.00** | **100.00** | 97.00 | 98.00 |
| InfoE | YieldE | F1 | 60.00 | 60.00 | 62.00 | 57.00 | 56.00 | 60.00 | 60.00 | 56.00 | **79.00** |
| InducGen | TopC | Accuracy | 40.00 | 50.00 | 48.00 | 47.00 | 43.00 | 54.00 | 49.00 | 56.00 | 30.00 |
| InducGen | ReactTR | F1 | 60.00 | 60.00 | 71.00 | 44.00 | 40.00 | 69.00 | 46.00 | 61.00 | 67.00 |
| *Molecular Understanding* | | | | | | | | | | | |
| MNGen | MolNG | Tanimoto (valid) | 51.04 (78%) | 54.56 (80%) | 41.57 (90%) | 31.43 (77%) | 38.25 (80%) | 53.15 (90%) | 48.84 (96%) | 25.18 (77%) | 18.34 (75%) |
| MNTrans | IUPAC2MF | L2 | 0.6632 | 0.7636 | 0.4944 | 0.3563 | 0.3847 | 0.6303 | 0.5908 | 0.2795 | 0.1652 |
| MNTrans | SMILES2MF | L2 | 0.5833 | 0.5942 | 0.2858 | 0.3233 | 0.3359 | 0.4569 | 0.3651 | 0.1953 | 0.2238 |
| MNTrans | IUPAC2SMILES | Tanimoto (valid) | 31.51 (52%) | 33.63 (52%) | 31.71 (83%) | 29.33 (65%) | 40.07 (75%) | 33.49 (67%) | 49.60 (88%) | 16.73 (65%) | 10.88 (60%) |
| MNTrans | SMILES2IUPAC | Exact Match | 0.00 | 0.00 | 0.00 | 0.40 | 0.40 | 1.20 | 0.00 | 0.00 | 0.00 |
| MNTrans | SMILES2IUPAC | BLEU | 3.44 | 4.49 | 1.37 | 4.19 | 4.49 | 4.33 | 2.53 | 1.00 | 0.11 |
| MNTrans | SMILES2IUPAC | Tanimoto | 0.00 | 0.00 | 12.69 | 17.03 | 21.01 | 24.25 | 17.86 | 13.05 | 7.42 |
| MNTrans | S2S | Tanimoto (valid) | 15.17 (44%) | 22.62 (80%) | 18.24 (74%) | 12.16 (72%) | 15.70 (68%) | 21.25 (85%) | 21.76 (62%) | 18.80 (72%) | 14.37 (79%) |
| MPP | MolPC | Accuracy | 73.08 | 71.60 | 68.55 | 63.23 | 58.49 | 66.72 | 55.79 | 56.87 | 58.71 |
| MPP | MolPR | NRMSE (valid) | 0.2574 (100%) | 0.2536 (100%) | 0.4128 (85%) | 3.3664 (98%) | 5.2053 (98%) | 0.2697 (100%) | 0.2934 (99%) | 0.3779 (98%) | 0.3860 (100%) |
| MolDesc | Mol2PC | LLM Score | 18.50 | 24.50 | 8.30 | 21.60 | 21.30 | 8.70 | 14.10 | 0.40 | 0.20 |
| *Scientific Knowledge Deduction* | | | | | | | | | | | |
| ReSyn | SubRec | F1 | 2.67 | 3.00 | 0.43 | 1.09 | 2.05 | 2.03 | 1.36 | 0.00 | 0.00 |
| ReSyn | PathRec | LLM Score | **52.50** | 40.63 | 25.00 | 29.25 | 28.75 | 33.13 | 24.00 | 33.38 | 41.13 |
| ReSyn | SynDE | NRMSE (valid) | 0.3806 (100%) | 0.5517 (100%) | 0.4856 (100%) | 0.7561 (100%) | 0.6454 (100%) | 0.5380 (100%) | 0.6527 (96%) | 0.3208 (100%) | 0.3251 (100%) |
| RCRec | LRec | F1 | 12.00 | 18.00 | 15.60 | 11.20 | 8.00 | 5.60 | 11.60 | 16.40 | 6.00 |
| RCRec | RRec | F1 | 45.00 | 41.67 | 21.31 | 32.33 | 33.65 | 30.54 | 12.39 | 37.26 | 35.27 |
| RCRec | SolvRec | F1 | 46.00 | 26.00 | 26.40 | 34.40 | 22.40 | 48.00 | 41.60 | 46.80 | 51.20 |
| RCRec | CatRec | F1 | 32.50 | 25.83 | 5.00 | 5.08 | 3.33 | 34.67 | 2.00 | 17.04 | 0 |
| RCRec | TempRec | NRMSE (valid) | 0.4951 (100%) | 0.4137 (100%) | 0.4841 (100%) | 0.3745 (100%) | 0.4625 (100%) | 0.4141 (100%) | 0.3170 (100%) | 0.4143 (100%) | 0.2561 (100%) |
| RCRec | TimeRec | NRMSE | 0.2071 (100%) | 0.1970 (100%) | 0.2164 (100%) | 0.1918 (100%) | 0.2614 (100%) | 0.1980 (100%) | 0.2085 (100%) | **0.1870** (100%) | 0.2080 (100%) |
| ROP | PPred | F1 | 12.00 | 20.00 | 1.07 | 11.87 | 16.19 | 14.10 | 0.63 | 0.40 | 0.96 |
| ROP | YPred | Accuracy | 54.00 | 34.00 | 48.50 | 75.00 | 32.50 | 40.50 | 40.50 | 61.00 | 88.00 |
| ROP | RatePred | Overlap | 16.74 | 14.41 | 20.27 | 17.17 | 15.82 | 19.24 | 13.45 | 15.82 | 15.40 |
| RMA | IMDer | LLM Score | 81.25 | 77.50 | 83.50 | 79.75 | 81.50 | 79.25 | **84.75** | 77.25 | 68.25 |

between the predicted and reference molecular formulas. A higher value indicates greater similarity between formulas.

- **Overlap:** Used to assess the proximity between the predicted range and the reference range. It is calculated as the length of the intersection divided by the length of the union of the predicted and reference ranges.

# D    FULL PERFORMANCE RESULTS

## D.1    PERFORMANCE RESULT OF 0-SHOT SETTINGS

The table 7 and the table 8 show the complete experiment results of all models under the zero-shot setting. We tested all the aforementioned models under zero-shot settings on ChemEval, as analyzed in Section 4.2.1. The results demonstrate that general-purpose models perform relatively well on knowledge question answering and literature comprehension tasks, while specialized models excel in more complex chemical tasks such as molecular property prediction. For certain tasks like CatRec, most models struggled to generate valid outputs, resulting in scores of zero.

## D.2    PERFORMANCE RESULT OF MULTIMODAL TASKS

The table 9 shows the performance of mainstream multimodal large language models on ChemEval's multimodal tasks, with '-' indicating meaningless responses. While most models handle basic tasks like molecular formula identification adequately, they struggle significantly with more complex challenges involving chemical reaction pathways and molecular properties. This performance gap widens further in Molecular Understanding and Scientific Reasoning tasks, which require both accurate molecular structure recognition from visual inputs and comprehensive chemical knowledge application. Our evaluation focused solely on general-purpose multimodal models, excluding chemistry-specific ones. As multimodal capabilities become increasingly essential in chemical research, this represents a critical area requiring urgent development.

Table 11: Performance overview of multi-level 3-shot text tasks on ChemEval (part 2).

| Dimension | Task | Metric | Qwen2.5-14B | Qwen2.5-7B | Llama3.3-8B | Grok3 | Gemini-2.5-Pro | ChemDFM | ChemLLM | LlaSMol | ChemSpark |
|---|---|---|---|---|---|---|---|---|---|---|---|
| *Advanced Knowledge Question Answering* | | | | | | | | | | | |
| ObjQA | MCTask | Accuracy | 64.80 | 55.60 | 38.40 | 70.40 | **90.80** | 44.80 | 13.60 | 4.00 | 32.00 |
| ObjQA | FBTask | LLM Score | 41.00 | 34.35 | 29.68 | 49.19 | 56.66 | 20.98 | 55.40 | 29.28 | 26.20 |
| ObjQA | TFTask | Accuracy | 61.60 | 63.60 | 46.80 | 74.40 | 72.00 | 65.20 | 0.80 | 38.00 | 57.20 |
| SubjQA | SATask | LLM Score | 52.20 | 48.70 | 29.00 | **73.00** | 70.00 | 30.50 | 11.50 | 23.50 | 31.60 |
| SubjQA | CalcTask | LLM Score | 51.10 | 40.80 | 19.70 | 79.30 | **81.60** | 16.40 | 35.46 | 68.37 | 15.80 |
| *Literature Understanding* | | | | | | | | | | | |
| InfoE | CNER | F1 | 57.42 | 64.84 | 51.35 | 61.47 | **73.62** | 36.98 | 0.09 | 9.04 | 72.30 |
| InfoE | CERC | F1 | 26.59 | 25.42 | 15.34 | 28.66 | 29.69 | 0.37 | 0.28 | 0.00 | **37.18** |
| InfoE | SubE | Accuracy | 62.69 | 68.17 | 57.71 | 79.42 | 76.29 | 20.04 | 0.00 | 0.00 | 72.86 |
| InfoE | AddE | F1 | 92.33 | 53.24 | 41.71 | 92.66 | **95.00** | 47.13 | 0.29 | 0.00 | 67.00 |
| InfoE | SolvE | F1 | 83.50 | 74.00 | 69.00 | 81.00 | 84.67 | 71.25 | 0.43 | 0.05 | **85.23** |
| InfoE | TempE | F1 | 70.00 | 79.00 | 69.00 | 79.00 | 77.00 | 41.00 | 1.53 | 0.00 | 80.00 |
| InfoE | TimeE | F1 | 95.00 | 89.00 | 89.00 | 95.00 | 95.00 | 78.00 | 0.98 | 0.00 | **95.00** |
| InfoE | ProdE | Accuracy | 84.55 | 83.14 | 73.26 | 90.62 | 93.75 | 8.83 | 0.00 | 0.00 | **98.40** |
| InfoE | CharME | F1 | 70.25 | 62.96 | 32.72 | 79.36 | **80.09** | 17.83 | 0.00 | 0.00 | 39.12 |
| InfoE | CatTE | F1 | 82.00 | 78.00 | 71.00 | **100.00** | 99.00 | 44.00 | 0.00 | 0.00 | 26.00 |
| InfoE | YieldE | F1 | 69.00 | 60.00 | 61.00 | 55.00 | 59.50 | 41.00 | 0.00 | 0.00 | 69.00 |
| InducGen | TopC | Accuracy | 49.00 | 47.00 | 28.00 | 46.00 | **73.00** | 27.00 | 0.00 | 0.00 | 25.00 |
| InducGen | ReactTR | F1 | 48.00 | 40.00 | 39.00 | **79.00** | 59.00 | 26.00 | 0.00 | 0.00 | 32.00 |
| *Molecular Understanding* | | | | | | | | | | | |
| MNGen | MolNG | Tanimoto (valid) | 10.27 (55%) | 4.71 (36%) | 7.51 (34%) | 49.26 (92%) | **72.33 (92%)** | 34.29 (69%) | 0.00 (0%) | 0.00 (0%) | 61.38 (95%) |
| MNTrans | IUPAC2MF | L2 | 0.1864 | 0.1719 | 0.2619 | 0.3393 | **0.8294** | 0.3225 | 0.0102 | 0.0000 | 0.8176 |
| MNTrans | SMILES2MF | L2 | 0.1333 | 0.1360 | 0.1674 | 0.1674 | | 0.4025 | 0.0072 | 0.0054 | **0.7224** |
| MNTrans | IUPAC2SMILES | Tanimoto (valid) | 7.67 (48%) | 3.51 (30%) | 2.37 (14%) | 65.15 (94%) | 59.44 (87%) | 38.66 (88%) | 0.00 (0%) | 0.00 (0%) | **83.98 (99%)** |
| MNTrans | SMILES2IUPAC | Exact Match | 0.00 | 0.00 | 0.00 | 0.00 | 0.40 | 0.00 | 0.00 | 0.00 | **10.80** |
| MNTrans | SMILES2IUPAC | BLEU | 0.62 | 0.15 | 0.13 | 3.44 | 13.61 | 0.26 | 0.08 | 0.00 | **45.96** |
| MNTrans | SMILES2IUPAC | Tanimoto | 7.80 | 3.39 | 1.91 | 54.63 | 1.82 | 0.00 | 0.00 | 0.00 | **61.08** |
| MNTrans | S2S | Tanimoto (valid) | 12.19 (71%) | 6.28 (56%) | 3.51 (47%) | 27.58 (87%) | 20.11 (74%) | 0.94 (25%) | 0.00 (0%) | 0.00 (2%) | **79.68 (89%)** |
| MPP | MolPC | Accuracy | 66.84 | 59.77 | 53.20 | 61.71 | 67.62 | 56.65 | 0.00 | 40.00 | **82.88** |
| MPP | MolPR | NRMSE (valid) | 1.6757 (100%) | 0.5915 (100%) | 50.9659 (81%) | 0.2886 (100%) | **0.2213 (100%)** | 1.6438 (87%) | 8.2422 (98%) | 10.0340 (89%) | 1.1634 (100%) |
| MolDesc | Mol2PC | LLM Score | 2.40 | 1.90 | 7.80 | 24.40 | 2.30 | 0.00 | 0.00 | 9.50 | **66.20** |
| *Scientific Knowledge Deduction* | | | | | | | | | | | |
| ReSyn | SubRec | F1 | 0.20 | 0.20 | 0.00 | 0.80 | 0.00 | 2.74 | 0.00 | 0.00 | **10.45** |
| ReSyn | PathRec | LLM Score | 28.75 | 23.50 | 17.88 | 25.25 | 43.00 | 28.75 | 6.75 | 17.50 | 27.00 |
| ReSyn | SynDE | NRMSE (100%) | 0.3223 (100%) | 0.4794 (100%) | 0.7969 (100%) | **0.2716 (100%)** | 0.4284 (100%) | 0.6243 (51%) | 0.6246 (100%) | 0.4367 (95%) | 0.5968 (66%) |
| RCRec | LRec | F1 | 9.20 | 6.40 | 2.40 | **29.60** | 0.00 | 12.49 | 5.60 | 0.00 | 16.80 |
| RCRec | RRec | F1 | 41.69 | 30.28 | 30.00 | 35.14 | 1.87 | 14.21 | 5.60 | 0.00 | **57.45** |
| RCRec | SolvRec | F1 | 26.00 | 48.00 | 33.80 | 30.40 | 0.00 | 24.59 | 0.00 | 0.00 | 32.00 |
| RCRec | CatRec | F1 | 18.67 | 8.13 | 0.25 | 2.89 | 2.89 | 3.90 | 3.43 | 1.80 | 1.97 |
| RCRec | TempRec | NRMSE (valid) | 0.5359 (100%) | 0.4211 (100%) | 0.7066 (89%) | 0.1687 (100%) | **0.1479 (100%)** | 0.6583 (99%) | 1.0526 (100%) | 0.9240 (90%) | 0.2682 (100%) |
| RCRec | TimeRec | NRMSE | 0.2053 (100%) | 0.2053 (100%) | 0.9478 (100%) | 0.1944 (100%) | 0.2090 (100%) | 0.1970 (100%) | 0.4404 (100%) | 0.3085 (100%) | 0.4021 (100%) |
| ROP | PPred | F1 | 0.00 | 0.40 | 0.00 | 10.87 | 30.00 | 11.93 | 0.00 | 0.00 | **53.60** |
| ROP | YPred | Accuracy | 92.00 | 92.00 | 22.00 | 9.50 | 33.00 | 36.80 | 0.00 | 0.00 | **88.50** |
| ROP | RatePred | Overlap | 16.71 | 12.29 | 14.29 | 22.83 | **29.08** | 17.46 | 0.00 | 0.00 | 11.03 |
| RMA | IMDer | LLM Score | 74.25 | 25.25 | 67.50 | 80.50 | 83.00 | 42.25 | 4.75 | 3.75 | 73.25 |

Table 12: Standard deviation across five trials for different models on ChemEval. Claude3.7T denotes Claude 3.7-Sonnet-Thinking, whereas Claude3.7N denotes Claude 3.7-Sonnet.

| Task Metric | SATask LLM Score | CalcTask LLM Score | CNER F1 | CERC F1 | ProdE Accuracy | S2S Tanimoto | MolPC Accuracy | LRec F1 | PPred F1 |
|---|---|---|---|---|---|---|---|---|---|
| GPT-4o | 61.20 ± 2.25 | 61.80 ± 1.21 | 65.76 ± 1.58 | 25.66 ± 1.48 | 86.09 ± 1.45 | 13.41 ± 1.39 | 64.57 ± 1.23 | 13.20 ± 2.99 | 1.67 ± 1.52 |
| claude3.7T | 56.70 ± 1.81 | 55.74 ± 2.82 | 60.21 ± 2.02 | 25.19 ± 1.91 | 82.39 ± 2.53 | 9.37 ± 0.78 | 58.90 ± 1.96 | 2.00 ± 1.26 | 12.27 ± 4.71 |
| claude3.7N | 55.10 ± 2.18 | 53.60 ± 2.15 | 54.55 ± 4.02 | 24.77 ± 1.18 | 85.04 ± 1.88 | 10.58 ± 1.14 | 54.37 ± 3.24 | 4.40 ± 1.50 | 16.16 ± 1.89 |
| Deepseek-R1 | 68.50 ± 2.21 | 76.10 ± 2.40 | 64.14 ± 1.72 | 27.18 ± 0.44 | 91.20 ± 0.35 | 16.04 ± 1.12 | 53.55 ± 0.63 | 6.80 ± 2.04 | 11.97 ± 1.73 |
| Deepseek-V3 | 71.70 ± 1.91 | 79.20 ± 2.94 | 60.85 ± 1.13 | 24.94 ± 1.12 | 87.52 ± 2.56 | 16.27 ± 1.44 | 48.73 ± 1.43 | 7.60 ± 2.33 | 0.93 ± 1.14 |
| Qwen2.5-72B | 58.50 ± 2.24 | 61.90 ± 2.08 | 61.61 ± 0.81 | 26.05 ± 0.84 | 84.86 ± 1.15 | 11.47 ± 1.17 | 48.13 ± 0.65 | 4.40 ± 1.50 | 1.73 ± 1.50 |
| LLama3.3-8B | 38.40 ± 1.93 | 28.00 ± 0.95 | 55.34 ± 3.85 | 17.31 ± 2.31 | 74.54 ± 1.56 | 1.74 ± 0.65 | 47.26 ± 1.86 | 2.13 ± 1.29 | 0.00 ± 0.00 |
| Grok3 | 73.59 ± 1.16 | 81.20 ± 1.60 | 60.75 ± 0.34 | 26.04 ± 0.61 | 91.04 ± 0.28 | 17.56 ± 1.75 | 56.62 ± 0.76 | 36.00 ± 1.26 | 11.33 ± 1.54 |
| Gemini-2.5-Pro | 72.00 ± 1.41 | 82.40 ± 0.97 | 68.30 ± 0.99 | 25.43 ± 1.63 | 92.82 ± 1.92 | 13.13 ± 1.01 | 63.63 ± 1.10 | 0.00 ± 0.00 | 29.20 ± 6.01 |
| ChemDFM | 32.20 ± 1.57 | 14.70 ± 1.17 | 41.17 ± 2.25 | 8.74 ± 2.52 | 34.73 ± 2.94 | 2.12 ± 0.31 | 61.35 ± 0.80 | 26.00 ± 3.79 | 18.80 ± 2.29 |
| ChemLLM | 13.20 ± 1.03 | 15.90 ± 2.91 | 0.16 ± 0.32 | 0.24 ± 0.12 | 0.00 ± 0.00 | 0.00 ± 0.00 | 0.00 ± 0.00 | 0.00 ± 0.00 | 0.00 ± 0.00 |
| ChemSpark | 33.60 ± 0.97 | 18.50 ± 2.02 | 71.44 ± 1.13 | 39.27 ± 2.59 | 94.40 ± 0.23 | 87.36 ± 1.46 | 85.57 ± 2.19 | 37.60 ± 0.80 | 56.40 ± 3.44 |

## D.3 PERFORMANCE RESULT OF 3-SHOT SETTING

As shown in the table 10 and the table 11, we evaluated all the aforementioned models under 3-shot settings on ChemEval. The results indicate that, similar to the zero-shot scenario, general-purpose models perform relatively well on advanced knowledge question answering and literature understanding tasks, while struggling with more complex molecular understanding and scientific knowledge deduction tasks. Specialized models such as ChemLLM and LlaSMol, due to their poor instruction-following capabilities, failed to return meaningful responses for most tasks, resulting in anomalous scores. These findings corroborate our previous analysis.

## E RESULTS OF ANALYSIS EXPERIMENTS

We conducted experimental analyses in two key areas. First, to establish the reliability of ChemEval metrics and demonstrate our evaluation framework's robustness, we conducted three repeated trials across identical task categories and calculated the standard deviation of results. Due to computational resource limitations, we were unable to conduct comprehensive experiments on all models and tasks. Therefore, we selected representative models and tasks for evaluation. Second, we investi-

Table 13: Experimental Results on CoT and Format Constraints.

| Dimension | Task | Metric | ChemDFM-NoFormat | ChemDFM-CoT | ChemLLm-NoFormat | Llasmol-NoFormat | Qwen2.5-7B-CoT |
|---|---|---|---|---|---|---|---|
| *Advanced Knowledge Question Answering* | | | | | | | |
| ObjQA | MCTask | Accuracy | 36.00 ↓5.20 | 32.00 ↓9.20 | 28.00 ↑3.60 | 24.00 | 50.00 ↓9.60 |
| ObjQA | FBTask | LLM Score | 24.00 ↓0.16 | 25.38 ↑1.22 | 31.58 ↓3.39 | 20.88 ↑6.96 | 27.64 ↓11.88 |
| ObjQA | TFTask | Accuracy | 46.00 | 32.00 ↓14.00 | 16.00 ↓3.20 | 56.00 ↓2.00 | 70.00 ↑14.80 |
| SubjQA | SATask | LLM Score | 44.80 ↑12.60 | 44.40 ↑12.20 | 32.40 ↑19.20 | 30.00 ↑15.50 | 57.60 ↑6.80 |
| SubjQA | CalcTask | LLM Score | 32.00 ↑17.30 | 32.40 ↑16.50 | 32.40 ↑16.50 | 22.00 ↑14.50 | 51.60 ↑8.00 |
| *Literature Understanding* | | | | | | | |
| InfoE | CNER | F1 | 43.44 ↑2.27 | 37.98 ↓3.19 | 47.61 ↑47.45 | 1.00 ↓10.62 | 67.02 ↑5.75 |
| InfoE | CERC | F1 | 11.53 ↑2.79 | 9.69 ↑0.95 | 16.81 ↑16.57 | 4.13 ↑2.89 | 22.89 ↓3.21 |
| InfoE | SubE | Accuracy | 0.00 ↓20.07 | 0.00 ↓20.07 | 0.00 | 0.00 | 0.00 ↓58.43 |
| InfoE | AddE | F1 | 33.33 ↓11.67 | 46.67 ↑1.67 | 66.67 ↑66.67 | 36.67 ↑36.67 | 65.33 ↑3.66 |
| InfoE | SolvE | F1 | 65.00 ↓15.50 | 60.00 ↓20.50 | 76.50 ↑74.83 | 0.00 | 78.33 ↓4.17 |
| InfoE | TempE | F1 | 60.00 ↓14.33 | 70.00 ↓4.33 | 70.00 ↑66.77 | 40.00 ↑40.00 | 65.00 |
| InfoE | TimeE | F1 | 80.00 ↑2.00 | 90.00 ↑12.00 | 95.00 ↑92.69 | 50.00 ↑25.00 | 95.00 |
| InfoE | ProdE | Accuracy | 0.00 ↓34.73 | 0.61 ↓34.12 | 0.00 | 4.13 ↑4.13 | 26.51 ↓50.49 |
| InfoE | CharME | F1 | 74.96 ↑47.70 | 64.52 ↑37.26 | 65.00 ↑65.00 | 44.96 ↑44.96 | 65.38 ↑22.38 |
| InfoE | CatTE | F1 | 35.00 ↓14.00 | 40.00 ↓9.00 | 45.00 ↑45.00 | 0.00 ↓5.00 | 55.00 ↓9.00 |
| InfoE | YieldE | F1 | 60.00 ↑15.00 | 60.00 ↑15.00 | 55.00 ↑55.00 | 55.00 ↓50.00 | 50.00 ↓17.00 |
| InducGen | AbsGen | LLM Score | 20.00 ↑20.00 | 20.00 ↑20.00 | 20.00 ↑14.50 | 11.00 ↓15.25 | 73.00 ↑18.25 |
| InducGen | OLGen | LLM Score | 19.00 ↑19.00 | 18.00 ↑18.00 | 40.00 ↑36.25 | 25.00 ↓6.25 | 58.00 ↑30.25 |
| InducGen | TopC | Accuracy | 30.00 ↓21.00 | 45.00 ↓6.00 | 35.00 ↑35.00 | 20.00 ↑20.00 | 45.00 ↑4.00 |
| InducGen | ReactTR | F1 | 25.00 ↑12.00 | 15.00 ↑2.00 | 30.00 ↑30.00 | 0.00 ↓5.00 | 20.00 ↓11.00 |
| *Molecular Understanding* | | | | | | | |
| MNGen | MolNG | Tanimoto (valid) | 71.94 (94%) ↑24.88 | 61.03 (92%) ↑13.97 | 0.62 (2%) ↑0.62 | 0.0 (0%) ↓3.71 | 3.44 (26%) ↓0.48 |
| MNTrans | IUPAC2MF | L2 | 68.15 ↑6.96 | 21.15 ↓40.04 | 6.99 ↑2.45 | 1.00 ↑1.00 | 9.93 ↓8.63 |
| MNTrans | SMILES2MF | L2 | 61.27 ↓2.72 | 17.14 ↓46.85 | 4.23 ↑0.48 | 0.00 | 3.96 ↓5.84 |
| MNTrans | IUPAC2SMILES | Tanimoto (valid) | 50.37 (96%) ↑3.66 | 44.77 (84%) ↓1.94 | 0.0 (0%) | 0.0 (0%) ↓4.70 | 3.23 (28%) ↓0.23 |
| MNTrans | S2S | Tanimoto (valid) | 0.14 (50%) ↓1.98 | 3.53 (46%) ↑1.41 | 2 (4%) ↑2.00 | 0.0 (0%) ↓0.60 | 2 (2%) ↓0.28 |
| MPP | MolIPC | Accuracy | 63.68 ↑2.33 | 57.12 ↓4.23 | 45.36 ↑45.36 | 54.92 ↑8.42 | 45.60 ↓18.45 |
| MPP | MolPR | NRMSE | 11.88 ↑383.07 | 240.91 ↑154.03 | 0.56 ↓178.80 | 12.19 ↑17.78 | 46.98 ↓38.39 |
| MolDesc | Mol2PC | LLM Score | 28.40 ↑25.30 | 28.00 ↑24.90 | 20.40 ↑20.10 | 25.60 ↑25.60 | 30.40 ↑15.90 |
| *Scientific Knowledge Deduction* | | | | | | | |
| ReSyn | SubRec | F1 | 0.00 ↓3.99 | 0.00 ↓3.99 | 0.00 | 1.33 ↑1.33 | 0.00 ↓1.42 |
| ReSyn | PathRec | LLM Score | 48.00 ↑23.88 | 40.50 ↑16.38 | 24.00 ↑13.13 | 30.50 ↑20.50 | 47.00 ↑19.88 |
| RCRec | LRec | F1 | 4.00 ↓22.00 | 4.80 ↓21.20 | 0.00 | 0.00 | 6.00 ↑3.20 |
| RCRec | RRec | F1 | 8.00 ↓5.13 | 9.33 ↓3.80 | 22.00 ↑22.00 | 0.00 | 44.00 ↑27.07 |
| RCRec | SolvRec | F1 | 6.00 ↓4.53 | 14.00 ↑3.47 | 8.00 ↑8.00 | 8.00 ↑8.00 | 20.00 ↓5.60 |
| RCRec | TempRec | NRMSE (valid) | 0.421 (85%) ↓0.04 | 0.2681 (85%) ↑0.11 | 0.9821 (45%) ↑0.14 | 7.9004 (15%) ↓7.03 | 0.3174 (55%) |
| RCRec | TimeRec | NRMSE (valid) | 0.5337 (70%) ↓0.06 | 0.6024 (55%) ↓0.13 | 1.306 (25%) ↑0.49 | - (0%) | 0.4396 (100%) ↓0.12 |
| ROP | PPred | F1 | 4.00 ↓14.80 | 14.00 ↓4.80 | 0.00 | 8.00 ↓8.00 | 0.00 |
| ROP | YPred | Accuracy | 52.00 (50%) ↑44.80 | 72.00 (50%) ↑64.80 | 70.00 (50%) ↑70.00 | 10.00 (50%) ↓18.00 | 80.00 (50%) ↑13.00 |
| ROP | RatePred | Overlap | 3.20 ↓0.59 | 9.86 ↑6.07 | 0.00 | 0.00 ↓3.68 | 2.70 ↓10.65 |
| RMA | IMDer | LLM Score | 57.00 ↓19.00 | 55.00 ↓21.00 | 37.00 ↑32.25 | 32.00 ↑30.50 | 56.00 ↓22.75 |

gated the differential impact of reasoning-oriented and format-constraint instructions in prompts, examining how reasoning capabilities and instruction-following ability influence model performance on complex chemical tasks.

## E.1 BENCHMARK STABILITY ASSESSMENT

The table 12 shows the result of our repeated experiments. The results reveal that standard deviations across most metrics remain below 5.0, demonstrating consistent performance across multiple evaluations. This statistical stability confirms the robustness of our evaluation framework, ensuring reliable and reproducible assessments of system performance.

## E.2 ANALYSIS OF CoT AND FORMAT CONSTRAINTS

As illustrated in Table 13, we evaluate four models: ChemDFM, ChemLLM, LlasMol, and Qwen2.5-7B under varied prompt configurations. When format restrictions were removed from the prompts, ChemDFM and LlasMol showed improved performance on simpler chemical tasks but declined on more complex ones. In contrast, ChemLLM achieved substantial performance gains across most tasks after the removal of format restrictions. This finding highlights that the loss of instruction-following ability can critically undermine the practical usability of domain-specific models. With respect to reasoning-oriented instructions, CoT prompting produced inconsistent outcomes for ChemDFM, enhancing performance in certain tasks while reducing it in others. Notably, Qwen2.5-7B consistently exhibited performance deterioration under CoT conditions, suggesting that explicit reasoning mechanisms contribute little to performance improvements on chemical tasks.

## E.3 ANALYSIS OF CHEMISTRY-SPECIFIC LLMs AND GENERAL-PURPOSE LLMs

Chemistry-specific language models are typically developed by fine-tuning open-source foundation models with domain-specific corpora. For instance, ChemLLM is derived from InternLM2-

Base-7B, while ChemDFM builds upon LLAMA-13B. Although these models incorporate extensive chemistry datasets and supplement them with general-domain data to mitigate catastrophic forgetting, their relatively small parameter scales impose fundamental limitations, particularly in instruction-following ability and task generalization. In contrast, closed-source state-of-the-art systems such as Gemini-2.0 and GPT-4o benefit from both larger parameter counts and massive pre-training corpora that include substantial amounts of open-source chemistry data, enabling them to preserve general reasoning capacity while delivering strong performance on chemistry tasks.

To further investigate the trade-offs between domain specialization and general-purpose robustness, we conducted a comparative study on Llasmol and its base model Mistral-7B. As shown in Table 14, the results reveal mixed outcomes: Llasmol achieves modest improvements on classification-based tasks such as multiple choice and true/false questions, but performs poorly on information extraction and molecular representation tasks where the base model significantly outperforms it. A closer analysis of the model outputs suggests that Llasmol suffers from weak instruction-following, often failing to generate answers in the specified format, as well as a tendency to produce irrelevant responses that are misaligned with the posed questions. These findings highlight the limitations of smaller domain-adapted models and underscore the advantages of large-scale general-purpose LLMs in achieving both reliability and relevance in specialized scientific applications.

# F  CASE STUDY

We conducted a detailed analysis of the models' outputs, systematically categorizing the most common types of errors they make, and provided two illustrative examples to highlight typical failure cases. These examples demonstrate the models' difficulties in accurately adhering to chemical nomenclature rules, predicting reaction substrates, and correctly interpreting molecular structures, thereby offering concrete insights into their limitations and areas for potential improvement.

In the Advanced Knowledge Question Answering and Literature Understanding levels, the model demonstrates strong proficiency in fundamental chemistry knowledge. Covering the four major branches of chemistry, the model shows some minor inaccuracies in understanding basic conceptual definitions in fill-in-the-blank tasks, resulting in occasional incorrect responses, though the overall error rate remains low. In short-answer tasks, the model provides detailed and accurate responses, reflecting its strengths in foundational chemistry question answering.

In the Molecular Understanding level, which primarily involves organic chemistry, the model exhibits errors related to unfamiliarity with nomenclature rules for natural products, heterocycles, and macrocycles, leading to name confusion; insufficient knowledge of special functional groups and substituents; and incomplete understanding of basic organic and stereochemical nomenclature rules, resulting in incorrect compound naming and misattribution of physicochemical properties.

In the Scientific Knowledge Deduction level, the model's errors include a lack of organic synthesis knowledge, insufficient understanding of organic reaction mechanisms, unfamiliarity with specific named reactions, limited grasp of reaction selectivity and reaction rules in organic compounds, and inaccurate prediction of target product structures.

These findings indicate that while the model performs well on foundational knowledge and basic molecular reasoning, it still has limitations in advanced organic synthesis reasoning, reaction condition recommendation, and product prediction, highlighting areas for further improvement.

Table 14: Comparison of LlaSMol and Mistral-7B performance on partial chemistry tasks.

| Tasks | Metric | Llasmol | Mistral-7B |
|---|---|---|---|
| Multiple Choice Task | Accuracy | 0.24 | 0.20 |
| True/False Task | Accuracy | 0.58 | 0.34 |
| Reaction Time Extraction | F1 | 0.25 | 0.90 |
| Reaction Temperature Extraction | F1 | 0.00 | 0.60 |
| IUPAC to Molecular Formula | L2 | 0.00 | 0.1142 |
| SMILES to Molecular Formula | L2 | 0.00 | 0.1076 |
| Intermediate Derivation | LLM score | 0.02 | 0.26 |

---

**Error case of Substrate Recommendation**

**Query**: Chemical reaction equations are typically expressed in the following form:

`reactant1.reactant2.reactant3...>>product.`

In this form, each substance (reactant/product) is represented using the SMILES notation. Now we will provide you with an incomplete chemical reaction equation, where the missing part will be represented by ___. The missing parts could be one or more substances. Based on the remaining portions of the reaction equation, please infer what the missing part could be. Please only provide the missing part in your response, without any additional content. The Incomplete equation is as follows:

`___>>CCOC(=O)C1C2=C(CCN1C(=O)C1=CC=CC(C(F)(F)F)=C1C1)N(C1=CC=CC=N1)C=N2.`

You must output your prediction, i.e. valid SMILES, and follow the output format exactly as follows: "answer":["valid SMILES 1", "valid SMILES 2",...]. I don't need any explanation, you just need to output your judgment in format.

**Groundtruth answer**:

`['CCOC(=O)C1NCCC2=C1N=CN2C1=CC=CC=N1', 'O=C(O)C1=CC=CC(C(F)(F)F)=C1C1']`

**Model answer**:

`{"answer":["CCOC(=O)C1C2=C(CCN1C(=O)C1=CC=CC=C1)N(C1=CC=CC=N1)C=N2", "C1C1=CC=C(C(F)(F)F)c=C1"]}`

**Error Analysis**: The model-predicted reaction substrate lacks the requisite reactive sites, preventing the two reactants from undergoing the reaction to yield the target product. Additionally, the model demonstrates inaccurate understanding of the target product's structural characteristics, and the reaction substrate contains extraneous functional groups.

---

**Error case of SMILES-to-IUPAC Translation**

**Query**: please give the IUPAC name of the molecule given by SMILES:

`CC(C)OC1=CC=C(C=C1)/C(=C\2/[C@@H](N(C(=0)C2=0)CCN3CCNCC3)C4=CC=C(C=C4)C1)/0`

You must output your prediction, i.e. a valid IUPAC, and follow the output format exactly as follows: "answer": "The answer you judge". I don't need any explanation, you just need to output your judgment information.

**Groundtruth answer**:

`(4E,5S)-5-(4-chlorophenyl)-4-[hydroxy-(4-propan-2-yloxyphenyl)methylidene]-1-(2-piperazin-1-ylethyl)pyrrolidine-2,3-dione`

**Model answer**:

`{"answer":["CCOC(=O)C1C2=C(CCN1C(=O)C1=CC=CC=C1)N(C1=CC=CC=N1)C=N2", "C1C1=CC=C(C(F)(F)F)c=C1"]}`

**Error analysis**: The molecular structure does not contain 4-hydroxypiperidinyl or 2-propenyl moieties, and the designation as a 1-one is also erroneous. The model exhibits an inadequate understanding of IUPAC nomenclature rules for heterocyclic compounds; the heterocycles actually present in this structure are pyrrole and piperazine rings. Furthermore, additional nomenclature errors include: incorrect enumeration of principal functional groups (the structure contains two ketone carbonyls), improper substituent naming, and incomplete stereochemical specification—while the structure possesses two stereocenters, the nomenclature designates only one of them.

---

## G   LLM USAGE

In this work, large language models were employed as a general assistive tool to improve the clarity and readability of the paper. Specifically, the models were used to polish grammar, punctuation, and phrasing in the text. No LLMs were used to generate original scientific ideas, analyze data, or draw conclusions; all scientific content, experimental design, analysis, and interpretation were entirely performed by the authors.

