# OpenReview forum: "ChemEval: A Multi-level and Fine-grained Chemical Capability Evaluation for Large Language Models"
_ICLR.cc/2026/Conference — ICLR 2026 Poster_

### Official Review · Reviewer_Rxp7 · 2025-10-28

**Soundness:** 4
**Presentation:** 2
**Contribution:** 3
**Rating:** 6
**Confidence:** 4

**Summary:**

This work proposes a comprehensive benchmark to evaluate the chemical capabilities of LLMs from multiple dimensions. Their benchmark, ChemEval, includes 4 progressive levels, evaluates 13 dimensions of LLMs’ capabilities, and features 62 distinct chemical tasks. The paper provides a thorough description of the data collection process and the task construction methodology. Finally, multiple advanced LLMs, including general LLMs and Chemical LLMs, are evaluated using ChemEval, and detailed analyses are conducted.

**Strengths:**

The paper demonstrates substantial effort in data collection and task construction. I highly recognize its important contribution to establishing a standardized benchmark for evaluating LLMs’ chemical capabilities and to enriching the diversity of evaluation tasks in this area.

**Weaknesses:**

1. The position of the figures should be organized more properly.
2. After carefully examining the authors’ anonymously released dataset, I found that the quality of some tasks is concerning. Certain tasks appear to lack chemical validity and, therefore, fail to reflect the model’s true chemical capability accurately. For example:
    * In the reaction time recommendation task (TimeRec), the reaction time should depend on factors such as catalyst type, reaction temperature, reactant concentration, and target yield. However, the task only provides the reactants and products, making the question essentially unanswerable.
    * The Synthetic Difficulty Evaluation task (SynDE) asks the model to assign a score to represent the synthetic difficulty of a molecule. The concept of “synthetic difficulty score” itself is human-defined, but the task neither specifies the range or meaning of the score nor clarifies whether a higher score indicates easier or harder synthesis. As a result, the question again becomes ill-posed and unanswerable.

    Issues like these raise serious concerns about the overall quality and validity of the dataset’s tasks. While expanding and diversifying the evaluation of chemical capabilities is indeed very important, the added tasks must at least ensure that they meaningfully and correctly assess a model’s chemical understanding. (In addition, the released dataset is difficult to inspect: the task categorizations are inconsistent with those described in the paper, many entries contain Chinese characters, and the number of samples differs from what is reported. These inconsistencies made it quite challenging to locate and examine data for specific tasks. I hope the authors will address these issues in future updates, as doing so would greatly improve the usability of the dataset.)
3. It seems that the number of entries in ChemEval is relatively small. For most individual tasks, the dataset contains fewer than 50 samples. It is unclear how such a limited number of instances can provide sufficient representativeness to reliably assess the model’s performance and capabilities on each task.
4. The conclusion regarding catastrophic forgetting in the paper is not properly reached. To properly evaluate whether catastrophic forgetting occurs, the comparison should at least be made between each chemistry LLM and its corresponding general LLM base model, such as ChemDFM vs. LLaMA, ChemLLM vs. InternLM, and LLasMol vs. Mistral. The fact that these chemical LLMs exhibit weaker general language performance than more advanced general LLMs does not provide valid evidence for the presence or absence of catastrophic forgetting during training.
5. A completely new model named ChemSpark appears in the paper, but there is no corresponding citation, paper, or webpage reference. I was unable to find any information about this model, so it is unclear how it was trained, whether there was any data leakage, or whether dedicated multi-task finetuning was involved. Without further clarification, the conclusions drawn from comparisons involving this model seem unreliable.

**Questions:**

Please refer to Weaknesses.

---

> ### Author Response · Authors · 2025-11-19
>
> We thank you for their thorough feedback and offer the following clarifications to your concerns.
>
> > W1: Figure Placement
>
> Thank you for the suggestion. We have adjusted the position of the figures accordingly.
>
> > W2: Clarification on TimeRec, SynDE Tasks, and Dataset Inspection
>
> We appreciate the reviewer’s insightful comments. Our clarifications regarding the TimeRec and SynDE tasks are as follows:
>
> **Regarding Reaction Time Recommendation (TimeRec):**
>
> We agree that reaction time is influenced by multiple factors such as catalysts, solvents, temperature, reactant concentration, and target yield. However, these conditions are frequently missing or difficult to standardize in real-world datasets. Therefore, we adopted a “standard condition assumption,” i.e., reactions of the same type are assumed to occur under similar typical laboratory conditions, making the structural features of reactants and products the primary variables that determine reaction time. We acknowledge the limitations of this simplification and will refine this task in future updates.
>
> **Regarding Synthetic Difficulty Evaluation (SynDE):**
>
> We apologize for the lack of clarity in the original manuscript. This task does not use an arbitrary human-defined score. Instead, it is based on the widely recognized Synthetic Accessibility Score (SAscore) [1] used in cheminformatics. SAscore ranges from 1 (very easy to synthesize) to 10 (very difficult to synthesize) and is computed using: Molecular complexity descriptors (e.g., chiral centers, ring systems), and Fragment contributions derived from millions of molecules in PubChem (“historical synthetic knowledge”). We have updated the manuscript to explicitly include the full definition of SAscore and its primary reference, ensuring that the evaluation criteria for this task are well-defined and scientifically grounded.
>
> [1] Ertl, Peter, and Ansgar Schuffenhauer. Estimation of synthetic accessibility score of drug-like molecules based on molecular complexity and fragment contributions. J. Cheminformatics 1.1 (2009): 8.
>
> **Regarding the reviewer’s concern about difficulty inspecting the anonymously released dataset:**
>
> We have rechecked the release, and the sample counts are consistent with those reported in the paper. The misunderstanding may have arisen because some task names in the anonymized dataset were presented in Chinese rather than aligned with the English names used in the manuscript. We apologize for this confusion. We have updated the anonymized public dataset to correct this issue. It should be noted that the AbsGen and OLGen tasks do not include 3-shot data due to the excessive length of the in-context examples. This missing portion of 3-shot data totals 40 entries. Consequently, while the 0-shot data totals 1,960 entries, the available 3-shot data is only 1,920 entries, resulting in a comprehensive publicly released dataset containing a total of 3,880 text entries.
>
> > W3: Clarification on Benchmark Variance and Representativeness
>
> Every item in the benchmark has undergone rigorous expert review and filtering to ensure that only high-difficulty, chemically rigorous “challenge problems” remain. We believe that performance on carefully designed, expert-curated items provides a reliable signal of a model’s true “chemical capability.” In Appendix E.1, we also provide standard deviation analyses across representative tasks and models. The results show relatively low variance, indicating that our benchmark yields stable and representative evaluations. We will continue to expand both the task set and sample size in future iterations.
>
> > W4: Analysis of Catastrophic Forgetting
>
> In Appendix E.3, we compare LlasMol with its base model Mistral across representative tasks at each level. These results support our analysis regarding catastrophic forgetting. Due to computational constraints, we plan to conduct a more comprehensive and systematic analysis in future work.
>
> > W5: Citation for the "ChemSpark" Model
>
> We apologize for the omission. The model referred to as “ChemSpark” in the manuscript corresponds to Spark-Chemistry-X1-13B [2], a specialized chemistry-oriented model developed by iFlyTek and released on the ModelScope platform. The model is fine-tuned from the iFLYTEK Spark-X1 base model on multiple chemistry-related datasets. We have added the official reference and model description to the revised manuscript to address your concerns.
>
> [2] https://www.modelscope.cn/models/iflytek/Spark-Chemistry-X1-13B
>
> We hope these responses and the corresponding revisions to the manuscript adequately address your concerns and help improve your evaluation of our work.

---

> ### Comment · Reviewer_Rxp7 · 2025-11-27
>
> Thank you for your detailed response. It clearly addresses Weakness 1 and Weakness 5 that I raised previously. However, I still have several concerns regarding the remaining points, and therefore, I have decided to maintain my current score for now. These concerns are as follows:
>
> **Regarding W2.**
>
> First, concerning the simplification of TimeRec: this simplification merely unifies the standard during evaluation, which does not seem to help produce a fairer assessment of model capabilities. Since the evaluated models are unaware of this standardized rule, and the evaluation does not explicitly communicate this rule through instruction (in fact, such inherently vague information is difficult to convey clearly through language), I believe this task still fails to genuinely reflect a model’s true ability to solve these types of chemical problems. In my view, an incorrect evaluation is more problematic than having no evaluation at all. Therefore, I hope you can better justify the rigor of this evaluation or the necessity of retaining this task.
>
> Second, regarding the SynDE task, thank you for introducing SAscore to me. However, I believe the core issue remains whether this task can truly reflect a model’s chemical understanding. In Table 1 of your paper, it is observed that most models fail to achieve meaningful scores on this task. I believe this is largely because these models do not possess prior knowledge of this SAscore system, which I do not think should be a required capability. What actually matters is the model’s ability to perceive the intrinsic difficulty of reactions, but not the SAscore standard. In my opinion, a good way to eliminate this confounding factor would be to reformulate the task into a comparison or selection task. I would be happy to discuss this further.
>
> In addition, I noticed that you rely heavily on expert annotation during data construction, yet the manuscript does not specify the number of experts involved or their level of expertise in each step. I hope you can provide this information.
>
> **Regarding W3.**
>
> Thank you for your explanation, but it does not resolve my concern. First, task difficulty alone does not demonstrate representativeness. For example, testing students with a series of difficult multiplication questions does not allow us to claim that the test reflects their overall arithmetic ability. The standard deviation you report also does not achieve this goal, as it primarily shows that model performance is stable across repeated sampling on those particular questions, but not the representativeness of the questions.
>
> However, I fully understand the difficulty and labor involved in data collection, as well as the challenge of achieving strong representativeness. Therefore, this issue does not negatively affect my scoring. However, I still hope you can provide more information to help readers better understand or estimate the representativeness or coverage of the questions, especially given that most data rely on subjective expert annotation. Experts’ domain distributions and personal biases could significantly affect the final data distribution.
>
> **Regarding W4.**
>
> I still believe that the current comparison is insufficient. You only compare part of the tasks on a single group of models, which does not seem adequate. In particular, considering that LlaSMol itself suffers from task generalization issues and behaves more like a multi-task specialist rather than a generalist model, catastrophic forgetting when facing tasks outside its training distribution is almost expected. While I agree that this statement is likely to be correct, a claim made in a research paper should be supported by more rigorous experimental evidence.
>
> These are my remaining concerns, and I welcome further discussions.

---

> ### Author Response · Authors · 2025-11-28
>
> We appreciate your response and the opportunity for further discussion. Below, we provide further clarifications regarding the concerns you raised.
>
> > W2:
>
> Regarding the evaluation metric for TimeRec:
>
> Thank you for pointing this out. We are actively considering revising the evaluation method to use an Overlap metric, similar to the RatePred task. With this approach, for specific types of reactions, a recommendation provided by the model that falls within a reasonable range would be considered correct. This metric not only assesses the model's understanding of common scenarios for such reactions but also eliminates the constraints imposed by rigid standardization rules.
>
> Regarding the SynDE task:
>
> Reformatting SynDE as a comparison or selection task is indeed a valuable suggestion. We agree that asking the model to compare reactions of varying difficulties would allow us to evaluate its understanding of synthesis difficulty without relying on specific scores. We plan to consider this modification for future work.
>
> However, we wish to clarify that the SAscore is a widely used metric in the chemistry domain for screening compounds based on synthesis difficulty, and we believe models should possess knowledge of such standard metrics. In our immediate revision, we will preliminarily modify the task to explicitly provide the model with the criteria for assessing reaction difficulty. In the future, we will consider further modifications, such as formulating it as a comparison task.
>
> Regarding the data standardization process:
>
> We have added detailed descriptions to the revised manuscript. Our data standardization follows a three-level workflow: "annotation → review → final verification".
>
> 1. Annotation: Performed by a team of 41 undergraduates majoring in chemistry who were trained using a Standard Operating Procedure manual.
>
> 2. Review: A team of 32 Master's students checked the data for consistency and correctness.
>
> 3. Final Verification: Three university chemistry professors conducted the final confirmation.
>
> > W3:
>
> We have also supplemented the description of our data construction process to address this. To enhance rigor and breadth, we collected extensive materials from domain experts, including: (1) a collection of approximately 500 university-level chemistry textbooks, workbooks, and exam papers; and (2) about 9,000 real experimental records provided by partner laboratories. The questions cover knowledge points required by university chemistry syllabi as well as frontline research scenarios, ensuring the distribution reflects the daily tasks of chemists.
>
> Subsequently, we manually constructed Q&A pairs based on these resources for specific task types. To prevent potential data leakage, we did not directly copy materials like textbook exercises. Instead, chemistry experts used them as references to compose entirely new questions based on target knowledge dimensions and task formats.
>
> We also analyzed the statistical distribution of chemical reaction types within the dataset. The proportions are as follows: Coupling reactions (62.4%), Substitution (15.8%), Redox (9.9%), Addition (4.7%), Elimination (2.5%), and Others (4.7%), where the 'Others' category includes nine types such as rearrangement, hydrolysis, and cyclization.
>
> We hope these clarifications address your concerns.

---

> ### Author Response · Authors · 2025-11-28
>
> > W4:
>
> We have conducted additional experiments. Due to computational resource constraints, we performed comparative experiments on representative tasks with two additional groups of models. The results are presented below:
>
> | Tasks | Metric | LLaMA | ChemDFM | InternLM | ChemLLM |
> | :--- | :--- | :--- | :--- | :--- | :--- |
> | Multiple Choice Task | Accuracy | 46.00 | 41.20 | 30.00 | 24.40 |
> | True/False Task | Accuracy | 46.00 | 46.00 | 56.00 | 19.20 |
> | Reaction Time Extraction | F1 | 75.00 | 78.00 | 65.00 | 23.10 |
> | Reaction Temperature Extraction | F1 | 70.00 | 74.33 | 55.00 | 3.23 |
> | IUPAC to Molecular Formula | L2 | 0.1326 | 0.6119 | 0.1344 |  0.0454 |
> | SMILES to Molecular Formula | L2 | 0.1056 | 0.6399 | 0.1014 | 0.0375 |
> | Intermediate Derivation | LLM score | 10.00 | 76.00 | 35.00 | 4.75 |
>
> The overall trend confirms our previous conclusions. In these new results, we observed that while ChemLLM and Llasmol possess chemical knowledge, their performance was hindered by limited instruction-following capabilities, often resulting in incomplete answers. In contrast, ChemDFM did not exhibit this issue; it effectively improved capabilities in chemical tasks, albeit with a slight performance drop in Level 1 tasks compared to general baselines. ChemLLM only integrates chemical corpora during the supervised fine-tuning phase; in contrast, ChemDFM is injected with large-scale chemical knowledge at the pre-training stage, which endows it with substantially stronger instruction-following performance on chemistry-specific tasks. This suggests that differences in training methods and data distributions may lead to varying manifestations of catastrophic forgetting across chemical models.
>
> We thank you for your meaningful comments and the discussion, which have enabled us to further improve our work. We hope our supplementary responses and revisions meet your expectations.

---

### Official Review · Reviewer_9BLh · 2025-10-31

**Soundness:** 3
**Presentation:** 3
**Contribution:** 3
**Rating:** 6
**Confidence:** 3

**Summary:**

The authers introduce ChemEval, a hierarchical benchmark for evaluating LLMs on chemical tasks. The framework comprises four progressive levels, 13 capability dimensions and 62 tasks cover a comprehensive chemistry domain task.
Evaluation of 16 general LLMs (GPT-4o, Claude-3.7, Gemini-2.5, Qwen, LLaMA, DeepSeek) and 4 chemistry-specific LLMs (ChemDFM, ChemLLM, LlaSMol, ChemSpark) reveals that general models excel at literature understanding and instruction-following but struggle with molecular reasoning, while specialized models show the opposite pattern, exhibiting catastrophic forgetting in general capabilities but superior performance in domain-specific tasks.

**Strengths:**

ChemEval's four-tier progression across 62 tasks with 37 novel dimensions (CharME, CatTE, multimodal spectroscopy/structure recognition) addresses gaps in prior benchmarks while covering real chemical workflows (retrosynthesis, reaction prediction, property estimation).

A comprehensive coverage of data: three-stage pipeline combining 25 open-source datasets with expert-curated materials, explicit deduplication against model training sets (ChemDFM, ChemLLM), and professional annotation validation ensures quality and reduces data leakage risks.

In this work, 20 models with domain-appropriate metrics (Tanimoto, NRMSE, validity ratios) was evaluated, quantifying the specialization-generalization tradeoff and providing actionable guidance for model development while open-source release enhances reproducibility.

**Weaknesses:**

Insufficient Justification for Task Selection and Missing Critical Tasks: The paper claims comprehensive coverage but still missing several core chemical AI tasks: Spectroscopy-to-structure elucidation, Safety/toxicity prediction. The 37 custom tasks lack clear selection criteria.

Single-domain focus: Evaluation is confined to chemistry; claims about LLM capabilities don't transfer to biology, materials science, or physics without cross-domain validation.

Minor:
Section 3.2.4 mentions "continuously adjusted instructions based on GPT-4o feedback" GPT-4o-generated prompts may bias evaluation toward GPT-4o's strengths. Were prompts validated by human experts?

The paper claims deduplication against ChemDFM/ChemLLM training sets, but what about GPT-4o's training data (cutoff Oct 2023)?

**Questions:**

What percentage of samples were rejected during filtering? What constituted "irrelevant" data (Figure 2 step b)? For LLM-as-judge metrics (LLM Score), which model was used—GPT-4o? This introduces potential bias where GPT-4o evaluates its own outputs.
Why does scaling fail? Table 3 shows Qwen2.5-7B→72B increases MCTask accuracy 59.6%→67.2% (+7.6pp) but decreases MolPC 64.04%→48.13% (-15.9pp). This is bizarre and unexplained. Is this overfitting? Requires error analysis.
Why do specialized models collapse on general tasks? The catastrophic forgetting observation (ChemLLM 0.00 on many tasks) is reported but not investigated.

---

> ### Author Response · Authors · 2025-11-19
>
> We appreciate the time and effort you invested in reviewing our work and offer the following clarifications to your queries.
>
> > W1: Task Inclusion (Spectrum/Toxicity) and Definition Criteria
>
> Regarding the two “missing” tasks you mentioned—spectrum-to-structure analysis and safety/toxicity prediction—we would like to clarify that these domains are already included in our benchmark:
>
> - Spectrum-related tasks: Under the Scientific Knowledge Deduction category, the “Molecular Property Description” task includes subtasks such as describing physicochemical properties from IR, Raman, and UV-Vis spectra, which cover spectrum-to-property reasoning.
>
> - Safety/toxicity prediction: Under Molecular Multi-modal Understanding, the “Molecular Property Prediction” task includes classification tasks such as SIDER and toxicity prediction.
>
> As for the criteria used to define our 37 custom tasks, our process was as follows:
>
> We first conducted an extensive literature review of chemical machine learning and chemical LLM research (e.g., [1–3]), and systematically summarized the core problem types emphasized in prior work. We then held multiple rounds of discussions with expert chemists to determine which tasks were both meaningful and feasible for benchmark construction. All 37 final tasks were derived from this process.
>
> [1] Irwin, R., Dimitriadis, S., He, J., Bjerrum, E.J., 2021. Chemformer: A Pre-Trained Transformer for Computational Chemistry.
>
> [2] Guo, Jiang, et al. Automated chemical reaction extraction from scientific literature. JCIM, 2021.
>
> [3] Zhang, Di, et al. ChemLLM: A Chemical Large Language Model. arXiv:2402.06852 (2024).
>
> > W2: Scope of the Benchmark
>
> Thank you for the suggestion. Our current work indeed focuses on evaluating large language models specifically within the chemistry domain, and our conclusions are correspondingly limited to chemistry and materials science. We will consider extending the evaluation to additional scientific domains in future work.
>
> > W3 & Q3: Prompt Verification and LLM Evaluator Robustness
>
> All prompts were manually verified by domain experts to avoid unintended bias from GPT-4o’s own generations. In addition, we conducted supplementary experiments in which ChatGLM-4.6 was used to evaluate GPT-4o’s outputs. The results were as follows:
>
> | Task | FBTask | SATask | CalcTask | AbsGen | OLGen | Mol2PC | PathRec | IMDer |
> | :--- | :--- | :--- | :--- | :--- | :--- | :--- | :--- | :--- |
> | GPT4o | 51.19 | 61.20 | 61.80 | 63.00 | 35.50 | 7.00 | 22.88 | 81.50 |
> | ChatGLM | 49.56 | 56.50 | 62.50 | 65.75 | 33.75 | 6.50 | 25.50 | 79.00 |
>
> The evaluations remain stable across different LLM judges, addressing concerns about potential evaluator bias. Going forward, we also plan to incorporate open-weight LLMs for scoring to ensure long-term reproducibility and sustained availability.
>
> > W4: Data Contamination Risks
>
> Because GPT-4o’s training corpus is not publicly accessible, direct decontamination is not possible. However, our data construction process inherently minimizes the risk of contamination:
>
> - Use of non-public data: A subset of our benchmark consists of internal laboratory experiment records that were never publicly released. Since these data were not available before GPT-4o’s training cutoff (Oct 2023), they cannot appear in its training set, guaranteeing absolute fairness for these tasks.
>
> - Use of official test splits: For tasks derived from open datasets, we strictly used the Test Split. Although we cannot audit GPT-4o’s full training data, using standardized test sets is the best accepted practice for contamination mitigation in current LLM evaluation.
>
> > Q1 & Q2: Definition of “irrelevant” data and Filtering Process
>
> In our filtering process, “irrelevant” is a general term referring to data unsuitable for inclusion in a high-quality benchmark. Human annotators with chemistry backgrounds manually reviewed these cases. They primarily included:
>
> - Task-mismatched: Samples inconsistent with the defined chemical task.
>
> - Ambiguous prompts: Questions with unclear wording that may lead to multiple interpretations.
>
> - Non-unique answers: Items with multiple plausible answers despite only one being annotated as correct.
>
> - Outdated knowledge: Items involving chemical knowledge that is no longer valid or has been revised.
>
> - Duplicates: Redundant or highly similar items identified through deduplication.
>
> Rejection proportion:
>
> Through this filtering process, we removed approximately 200 samples, mainly due to ambiguity, non-unique answers, or outdated knowledge. This represents roughly 2% of the initial dataset.
>
> We believe this filtering step is essential to ensuring the accuracy and fairness of the benchmark.

---

> ### Author Response · Authors · 2025-11-19
>
> > Q4: Analysis of 72B Model Overfitting and ChemLLM's "Instruction-Following Collapse"
>
> Analysis of 72B Model Overfitting: Thank you for raising this insightful point. We agree that the sharp performance drop in MolPC for the 72B model likely reflects a form of overfitting, rooted in a severe mismatch between model capacity and available domain-specific data. Although scaling laws typically predict improved performance with larger models given sufficient data, MolPC requires high-quality, highly specialized chemical data that are relatively limited in quantity. The 72B model’s capacity far exceeds what the domain dataset can support, resulting in degraded performance. In contrast, the 7B model has a more balanced capacity-to-data ratio, allowing it to generalize better on this specialized task and thus outperforming its larger counterpart.
>
> ChemLLM's "Instruction-Following Collapse": Regarding the catastrophic forgetting observed in ChemLLM, our analysis indicates that the decline is not merely traditional “knowledge forgetting,” in the sense of losing general-domain knowledge. Rather, the core issue is that after intensive domain-specific fine-tuning on chemistry tasks, the model experiences a severe degradation in its instruction-following capability for general-purpose prompts. In other words, even if the model “knows” the correct answer, it often fails to structure its output according to the formatting and response requirements of general-domain benchmarks (e.g., MMLU or MCTask), such as choosing among “A/B/C/D” options or producing answers in a required JSON schema. As a consequence, the model frequently produces irrelevant, off-format, or unparseable outputs, which are automatically scored as 0.00 under strict evaluation scripts. This phenomenon is fundamentally different from true semantic forgetting. Below we provide two concrete examples illustrating this behavior.
>
> Case:
>
> > Molecular Name-to-SMILES Generation — ChemLLM repeats the prompt and provides random text
>
> > Input:
>
> > A structured prompt requiring the model to generate a valid SMILES string corresponding to a complex amino tetrasaccharide, and to output the result in the exact format:
> {"answer": "a valid SMILES"}
>
> > ChemLLM Output:
>
> > ChemLLM again repeats the entire instruction, adds extraneous explanation, and finally outputs a nonsensical fragment resembling a corrupted SMILES-like string (e.g., CC(=O)N[C@@H]1COC([CH2+][S+](CCO[Si](#CN=C)[n-])=[NH3-O])...).
>
> > The model completely ignores the required output format and cannot be evaluated by automated scripts.
>
> We hope these clarifications and supplemental results strengthen the paper and adequately address the points you raised and help improve your evaluation of our work.

---

### Official Review · Reviewer_wAna · 2025-11-01

**Soundness:** 2
**Presentation:** 3
**Contribution:** 2
**Rating:** 4
**Confidence:** 4

**Summary:**

The paper proposes ChemEval, a hierarchical benchmark to evaluate LLMs’ chemical capabilities. It organizes 62 tasks into four progressive levels (from basic concepts and literature-style QA → extraction and structured understanding → molecular-level reasoning → high-level synthesis/mechanism tasks). The benchmark is mixed-source: some tasks are adapted from existing open datasets (Mol-Instructions, ChemLLM-Bench, SMolInstruct, Llasmol), and 37+ tasks are newly authored/curated by chemistry experts from collaborating universities, including proprietary laboratory records and multimodal materials (tables, spectra, reaction schemes).

**Strengths:**

1. Compared with prior suites (ChemLLM-Bench, ChemCrow eval, ChemBench-MM, Mol-Instructions), this work (i) unifies text-only and multimodal chemistry tasks in one place, (ii) explicitly stages difficulty into four levels, and (iii) evaluates many models under a comparable protocol.

2. Good task–metric alignment. Tasks that are structure-like or chemistry-like use appropriate metrics (Tanimoto, L2, exact-format scoring), while the knowledge / reading tasks use accuracy-style metrics.

**Weaknesses:**

1. The core test size is 3,120 items across 62 tasks → that’s ~50 items per task on average, sometimes less. For high-variance LLM outputs (esp. with many format-sensitive IE tasks), this is thin, and you can’t draw very strong conclusions about model scaling, few-shot gains, or thinking-vs-non-thinking from 30–80 examples. Yet the result section does exactly that. ICLR will push back: “you are positioning this as a comprehensive benchmark but the statistical mass is closer to a curated diagnostic suite.” The paper acknowledges the cost of expert construction, but the conclusion section still speaks as if we have a definitive benchmark. That’s overstated relative to the data volume

2. The paper does not adequately describe the annotation pipeline, especially for the scientific knowledge deduction tasks, which appear to be the most challenging and high-value part of ChemEval. In contrast to the relatively straightforward literature-understanding and molecule-level tasks, deduction requires multi-step reasoning, selection of relevant premises, and often domain-specific normalization. Right now, the description of how these instances were constructed, validated, and standardized is too brief to assess reliability. The authors are encouraged to spell out: (i) the source of the raw material (textbook, lab record, expert-written), (ii) how the gold answer was derived from it, (iii) whether multiple annotators or chemistry experts were involved, and (iv) how disagreements were resolved. Without this, it is hard to judge the difficulty and reproducibility of the most important part of the benchmark.

 Section 3.2.2–3.2.4 describes a 3-step pipeline (collect → filter → construct Q/A) but when we get to retrosynthesis, condition recommendation, reaction outcome prediction, mechanism analysis (the 13 tasks in Level 4), the paper stays at the narrative level.

3. Although the paper frames ChemEval as addressing scenarios where models must read heterogeneous chemical inputs (text, tables, schemes) and then reason, the experiments only evaluate plain LLM inference (mostly greedy decoding) without any agent-style or chemistry-tool–augmented setups. For this domain, that’s a real gap: in practice, chemistry LLMs are often run with RDKit/OPSIN/reaction-database calls or ChemCrow-style tool chains to normalize structures, check validity, and retrieve reaction conditions. Without at least a small ablation (e.g. LLM-only vs. LLM + RDKit name-to-structure vs. LLM + reaction lookup) on 2–3 tasks, it is hard to tell whether ChemEval can distinguish different modeling paradigms (language-only vs. tool-augmented) or whether it is mainly measuring prompt-following in a chemistry setting. As written, the results mostly re-confirm a known pattern (“frontier general models > small chemistry-tuned models”) but do not show that the benchmark is sensitive to the way models are used.

**Questions:**

Since many chemistry LLM applications are tool-augmented, can you report results for at least one LLM+tool pipeline (e.g., LLM → RDKit/OPSIN → answer) on a subset of tasks (name↔structure, condition extraction, reaction-type identification)?

id you consider evaluating a standard chemistry agent (e.g., ChemCrow-style tool-calling) as a reference system?

---

> ### Author Response · Authors · 2025-11-19
>
> We sincerely appreciate your detailed review and the insightful questions you've raised.
>
> > W1: Benchmark Size
>
> Thank you for your constructive feedback. Following your suggestion, we have refined parts of the manuscript wording to more accurately reflect the scope and contribution of our work.
>
> - High-difficulty: Although the number of samples in each task is relatively small, this is due to the fact that every single item underwent rigorous expert screening and verification, ensuring that only high-difficulty, chemically meaningful challenge problems were retained. We believe that performance on carefully designed, expert-curated, high-difficulty chemistry questions provides a strong and reliable signal of a model’s “chemical capability.”
>
> - Stability: Additionally, in Appendix E.1, we provide the standard deviation of evaluation scores across representative models on tasks at each level. The variance remains reasonably small, empirically proving that our current sample size is sufficient to yield stable signals for assessing model capabilities.
>
> We also plan to continuously expand both the dataset and the task set in future releases.
>
> > W2: the Annotation Pipeline and Data Sources
>
> Regarding your questions about the annotation pipeline, we provide the detailed clarification below:
>
> 1. Sources of Materials:
>
> Our raw materials come from two primary sources:
>
> (1) ~500 chemistry textbooks, problem sets, and university-level exam papers collected from multiple institutions;
>
> (2) ~9,000 authentic expert laboratory records provided by our collaborating research laboratories.
>
> 2. Annotation Method:
>
> - Advanced knowledge QA: Items from textbooks, exercises, or exams were not directly used as benchmark questions. Instead, they served only as conceptual references. New questions were written by chemistry experts based on the underlying knowledge dimensions to prevent potential data leakage.
>
> - Scientific knowledge deduction: Based on real experimental data and task settings, chemistry experts constructed both the questions and the corresponding answers.
>
> 3. Standard Answers:
>
> All questions and answers are grounded strictly in the factual content of the raw materials (e.g., textbook reference answers or empirical laboratory data). We emphasize that for “deduction” tasks, the answers are derived from factual or experimental evidence, not from annotator speculation.
>
> 4. Annotation Personnel:
>
> We employed a three-stage pipeline: annotation → review → final verification.
> First, trained chemistry undergraduates annotated the items according to a detailed SOP handbook;
> Second, graduate students checked consistency and correctness;
> Finally, experienced chemistry instructors conducted the terminal review and approval.
>
> 5. Conflict Resolution:
>
> We relied on a strict annotation manual and SOP to standardize all decisions. If ambiguity or non-uniqueness was discovered at any stage, the item was immediately removed from the benchmark to ensure objectivity and reproducibility.
>
> We hope these detailed explanations address your concerns. We have integrated these details into the revised Section 3.2 to provide a comprehensive description of our benchmark construction process.

---

> ### Author Response · Authors · 2025-11-19
>
> > W3 & Q: Supplemental Experiments on Tool-Augmented Pipelines
>
> In response to your request for evaluating tool-augmented pipelines, we conducted experiments using ChemCrow-public [1] with gpt-4-0613 as the LLM, assessing LLM+Tool performance on ChemEval. The results are shown below:
>
> | Dimension | Task | Metric | ChemCrow |
> | :--- | :--- | :--- | :--- |
> | ObjQA | MC Task | Accuracy | 58 |
> | ObjQA | FBTask | LLM Score | 43.14 |
> | ObjQA | TFBTask | Accuracy | 74 |
> | SubjQA | SATask | LLM Score | 43.5 |
> | SubjQA | CalcTask | LLM Score | 43.5 |
> | InfoE | CNER | F1 | 57.46 |
> | InfoE | CERC | F1 | 22.05 |
> | InfoE | SubE | Accuracy | 50.91 |
> | InfoE | AddE | F1 | 43.33 |
> | InfoE | SolvE | F1 | 87.5 |
> | InfoE | TempE | F1 | 65 |
> | InfoE | TimeE | F1 | 95 |
> | InfoE | ProdE | Accuracy | 71.38 |
> | InfoE | CharME | F1 | 25 |
> | InfoE | CatTE | F1 | 85 |
> | InfoE | YieldE | F1 | 50 |
> | InducGen | AbsGen | LLM Score | 57.5 |
> | InducGen | OLGen | LLM Score | 32.5 |
> | InducGen | TopC | Accuracy | 45 |
> | InducGen | ReactTR | F1 | 5 |
> | MNTrans | MolNG | Tanimoto (valid) | 40.92 (90%) |
> | MNTrans | IUPAC2MF | L2 | 0.1408 |
> | MNTrans | SMILES2MF | L2 | 0.3089 |
> | MNTrans | IUPAC2SMILES | Tanimoto (valid) | 25.68 (64%) |
> | MNTrans | SMILES2IUPAC | Exact Match | 0 |
> | MNTrans | SMILES2IUPAC | BLEU | 0.38 |
> | MNTrans | SMILES2IUPAC | Tanimoto | 0 |
> | MNTrans | SMILES2SELFIE | Tanimoto (valid) | 9.83 (38%) |
> | MPP | MolPC | Accuracy | 46 |
> | MPP | MolPR | NRMSE (valid) | 0.3408 (38%) |
> | MolDesc | Mol2PC | LLM Score | 21 |
> | ReSyn | SubRec | F1 | 0 |
> | ReSyn | PathRec | LLM Score | 48.75 |
> | ReSyn | SynDE | NRMSE (valid) | - (0%) |
> | RCRec | LRec | F1 | 18 |
> | RCRec | RRec | F1 | 36.65 |
> | RCRec | SolvRec | F1 | 12 |
> | RCRec | CatRec | F1 | 0 |
> | RCRec | TempRec | NRMSE (valid) | 0.2392 (85%) |
> | RCRec | TimeRec | NRMSE (valid) | 0.5209 (70%) |
> | ROP | PPred | F1 | 0 |
> | ROP | YPred | Accuracy | 24 |
> | ROP | RatePred | Overlap | 0 |
> | RMA | IMDer | LLM Score | 28.75 |
>
> Since ChemCrow relies on tool invocation, enforcing strict output-format constraints in the prompt often led to tool-call failures. Therefore, we removed these constraints during evaluation.
>
> From the results, we observe that ChemCrow’s overall performance is comparable to general-purpose LLMs. On certain tasks in Molecular Understanding and Scientific Knowledge Deduction, ChemCrow achieves performance close to that of specialized chemistry models, demonstrating the usefulness of domain tools. However, its performance remains low on some tasks. Possible reasons include:
>
> (1) The model's insufficient understanding of the chemistry problem prevents it from correctly planning the tools needed to solve it, resulting in incorrect or failed tool calls.
>
> (2) Several tool modules in the public ChemCrow implementation being unavailable or non-functional, which limits task completion.
> These observations highlight the importance of developing robust domain agents in chemistry, but also show that tool-augmented systems still rely on strong foundational chemical reasoning within the underlying model for effective task interpretation and planning.
>
> [1] https://github.com/ur-whitelab/chemcrow-public
>
> We hope these detailed explanations and additional results clarify our contributions and resolve your concerns and help improve your evaluation of our work.

---

### Official Review · Reviewer_DKh7 · 2025-11-02

**Soundness:** 3
**Presentation:** 3
**Contribution:** 3
**Rating:** 6
**Confidence:** 3

**Summary:**

the paper introduces ChemEval, a chemistry-specific evaluation suite designed with 4 progressive levels across 13 dimensions of LLM capabilities. The authors claim to innovatively introduce test sets related to information extraction. multimodal tasks are also included. Finally, tasks have been sourced from the web in openly accessible sites, as well as domain experts in chemistry.

**Strengths:**

The paper is well written and goes in depth into details of analyzing the data, evaluation metrics, and organizing the test-set into tasks, dimensions and levels, which makes the paper and the benchmark very valuable for the field.
The paper also gives a very detailed evaluation of several relevant llms such as closed-source state of the art models, as well as chemistry-specific models. The analysis is also in depth and discusses differences between e.g. reasoning and non-reasoning models, open and closed, and chemistry specific and non-specific.

**Weaknesses:**

Some of the tasks, admittedly due to their complexity, require evaluation with an LLM Score. This score in itself was not evaluated, benchmarked against human baselines, or revised thoroughly. Additionally the LLM used to score was not disclosed, which further hinders transparency and reproducibility.
Given these points, it is not clear how the authors plan to make this benchmark endure the test of time. If the model is an open-weights model it would be good to mention it and release the weights as an asset together with the benchmark, so that the community can run these evaluations. If the model is a proprietary one, it is even less clear how to do this, as the continued availability of api-based models is not guaranteed.
Furthermore, the paper is missing a discussion on why the specific llm used was used. For instance, what is the variance of the results if the LLM is changed? are there any specific differences in behaviour patterns between LLMs as evaluators?

It would also be great to provide a human baseline, to assess how good are expert chemists, and average humans, at the tasks in this evaluation set.

**Questions:**

Please provide more details and ablations on the LLM used for evaluation for some of the tasks. See weaknesses section for more details and elaborate on how do you plan to move about this. If this is to be truly open and sustainable over time, the evaluations should rely on a specific snapshot of an open-weights model.

---

> ### Author Response · Authors · 2025-11-19
>
> Thank you very much for your valuable and constructive feedback.
>
> > W1 & Q1: the Robustness and Transparency of LLM Evaluators
>
> Thank you for pointing this out. The LLM used for scoring in our experiments is GPT-4o-0806. In response to your concerns, we have added experiments evaluating the robustness of the scoring process using different LLM evaluators. Specifically, we used ChatGLM-4.6 to score the outputs generated by GPT-4o. The results are as follows:
>
> | Task | FBTask | SATask | CalcTask | AbsGen | OLGen | Mol2PC | PathRec | IMDer |
> | :--- | :--- | :--- | :--- | :--- | :--- | :--- | :--- | :--- |
> | GPT4o | 51.19 | 61.20 | 61.80 | 63.00 | 35.50 | 7.00 | 22.88 | 81.50 |
> | ChatGLM | 49.56 | 56.50 | 62.50 | 65.75 | 33.75 | 6.50 | 25.50 | 79.00 |
>
> As shown, the evaluation results remain relatively stable across different LLM scorers, indicating that our benchmark is not overly sensitive to a particular evaluator. This addresses the concern regarding potential variance or evaluator-specific behavioral patterns. Moving forward, we plan to incorporate open-weight LLMs as evaluators to further improve long-term accessibility and sustainability.
>
> > W2: Clarification on Adding Human Baseline Evaluations
>
> This is indeed a meaningful comparison, and we will consider adding such evaluations in future work. For non-experts with limited chemistry knowledge, such as individuals with only high-school–level chemistry background, the tasks in our benchmark are generally too specialized, and their performance would likely be close to random guessing. Therefore, while valuable, such human baselines require careful design and will be explored in subsequent extensions of the benchmark.
>
> We hope these clarifications adequately address your concerns and help improve your evaluation of our work.

---

### Author Response · Authors · 2025-11-30

# Global Response

We sincerely thank all the anonymous reviewers for their valuable and insightful comments. We have added some content to our revised paper according to the comments and questions. Here we provide the comprehensive final comment summarizing the reviewers' concerns:

### 1. **Evaluation Methodology and Robustness issues raised by reviewer DKh7 and reviewer 9BLh.**

Multiple reviewers questioned the robustness and transparency of LLM evaluators, expressing concerns about potential evaluator-specific bias and variance in scoring.

We conducted cross-evaluator experiments using ChatGLM-4.6 to score GPT-4o outputs, demonstrating stable evaluation results across different LLM judges. Besides, we also verified that all prompts were manually reviewed by domain experts to avoid bias.

### 2. **Benchmark Scale and Quality issues raised by reviewer wAna and reviewer Rxp7.**
Reviewer reviewer wAna questioned the relatively small sample size, while reviewer Rxp7 raised concerns about benchmark variance and representativeness.

We need to emphasize that every sample underwent rigorous expert screening, ensuring high-difficulty, chemically meaningful problems. We provided standard deviation analyses showing low variance across tasks and models, demonstrating that the current sample size yields stable evaluation signals.

### 3. **Data Construction and Annotation Process issues raised by reviewer wAna and reviewer 9BLh.**

Reviewer wAna and reviewer 9BLh requested detailed clarification about annotation pipelines, data sources, and quality control measures.

We provided comprehensive descriptions of their three-stage annotation process (annotation → review → final verification), involving 41 undergraduate chemistry majors, 32 graduate students, and 3 chemistry professors. We detailed data sources (~500 textbooks and ~9,000 laboratory records) and conflict resolution procedures.

### 4. **Task Design and Domain Coverage issues raised by reviewer 9BLh and reviewer Rxp7.**

Reviewer 9BLh questioned missing tasks (spectrum/toxicity analysis), while Reviewer  Rxp7 raised specific concerns about TimeRec and SynDE task definitions.

We clarified that the questioned tasks were already included in their benchmark under different categories. For TimeRec and SynDE, we provided detailed justifications and acknowledged limitations, proposing future improvements such as using overlap metrics and comparison-based evaluations.

### 5. **Data Contamination and Bias issues raised by reviewer 9BLh.**

Reviewer 9BLh expressed concerns about potential data contamination in GPT-4o training data.

We addressed this through two strategies: (1) using non-public laboratory experimental records that couldn't appear in GPT-4o's training set, and (2) strictly using official test splits from public datasets.

### 6. **Model Performance Analysis issues raised by reviewer 9BLh and reviewer Rxp7.**

Reviewer 9BLh and reviewer Rxp7 requested deeper analysis of unusual model behaviors, particularly 72B model overfitting and ChemLLM's instruction-following collapse.

We provided detailed technical explanations for these phenomena, distinguishing between capacity-data mismatch in large models and instruction-following degradation after domain-specific fine-tuning. And we conducted additional comparative experiments with multiple models to support their analysis.

### 7. **Tool-Augmented Systems and Baselines issues raised by reviewer wAna and reviewer DKh7.**
Reviewer wAna requested evaluation of tool-augmented pipelines, while reviewer DKh7 suggested human baseline comparisons.

We conducted experiments with ChemCrow-public, showing comparable performance to general LLMs but highlighting limitations in tool planning. For human baselines, we acknowledged the value but noted that non-expert performance would likely approach random guessing due to task complexity.

### 8. **Technical Issues and Reproducibility issues raised by reviewer Rxp7.**

Reviewer Rxp7 identified figure placement issues and dataset inspection difficulties.

We corrected figure positions and updated the anonymized dataset to fix naming inconsistencies between Chinese and English task names.

We have systematically addressed all major concerns through additional experiments, detailed explanations, and methodological improvements, demonstrating a commitment to benchmark quality and scientific rigor.

---

### Meta-Review · Area_Chair_Re3E · 2026-01-02

**Summary:**

The paper introduces a new benchmark for evaluating chemistry knowledge and reasoning abilities of LLMs. The benchmark focuses on introducing a greater diversity of tasks than prior work, including tasks beyond question and answering that require a greater diversity of evaluation metrics. Based on the proposed benchmark, the authors evaluate diverse general LLM and chemistry specific LLMs on the benchmark tasks and provide a variety of conclusions. This include a trade-off between general LLMs performing better and language conditioned tasks, such as QA, while chemistry specific LLMs perform better on more specialized chemistry tasks while degrading performance on language conditioned tasks. The analysis also includes details on Qwen models of different scales and the impact of reasoning models.

The reviewers generally praised the extensive set of benchmarking, along models and tasks, conducted by the paper as well as the extensiveness of tasks the benchmark introduces. Some reviewers also praise the inclusion of multimodal tasks and the alignment of diverse evaluation metrics with different tasks.

Most of the reviewers' concerns centered on providing greater transparency in how the benchmark was constructed, including the annotation process and the data sources. The authors provided additional details during the rebuttal with helpful details to that effect. Another concern from multiple reviewers related to the representativeness of the benchmark and stability of benchmark evaluation given the relative small size of question per task (~50). The authors' rebuttal provided additional details, pointing to a table in the appendix showing small variance and emphasizing difficulty and quality of the benchmark. While this partially addresses the concerns, the authors should acknowledge this as a limitation and adjust conclusions appropriately.

**Reviewer Concerns:**

Addressed Concerns:
* Reviewer DKh7's concern on using different LLMs as a judge was generally addressed.
* Reviewer wAna's concerns on benchmark annotation procedures and using tool-augmented LLMs were generally addressed.
* Reviewer 9BLh's concerns on scope of the benchmark and annotation procedures were mostly addressed. The authors also provided additional details on the overfitting case study and data contamination questions.
* Reviewer Rxp7's concerns on providing cleaner data and having greater details on annotation procedures were generally addressed.

Outstanding Concerns:
* Reviewer wAna's concerns on benchmark size and representativeness were partially addressed. The limitations pointed out should be acknowledged by the authors.
* Reviewer 9BLh's questions on task inclusions and definition were partially addressed.
* Reviewer Rxp7's concerns on benchmark representativeness were partially addressed. The concerns on the conclusions about catastrophic forgetting were also partially addressed.

**Reviewer Scores:**

* Reviewer DKh7's score remains at 6.
* Reviewer wAna's score increases to 6.
* Reviewer 9BLh's score remains at 6.
* Reviewer Rxp7's score remains at 6.

---

### Decision · Program_Chairs · 2026-01-26

Accept (Poster)